# Kolmogorov-Arnold Attention: Is Learnable Attention Better For Vision Transformers?

## Abstract

Kolmogorov-Arnold networks (KANs) are a remarkable innovation that consists of learnable activation functions, with the potential to capture more complex relationships from data. Presently, KANs are deployed by replacing multilayer perceptrons (MLPs) in deep networks, including advanced architectures such as vision Transformers (ViTs). Given the success of replacing MLP with KAN, this work asks whether KAN could learn token interactions. In this paper, we design the first learnable attention called **K**olmogorov-**Ar**nold **At**tention (KArAt) for ViTs that can operate on any basis, ranging from Fourier, Wavelets, Splines, to Rational Functions. However, learnable activations in the attention cause a memory explosion. To remedy this, we propose a modular version of KArAt that uses a low-rank approximation. By adopting the Fourier basis into this, Fourier-KArAt and its variants, in some cases, outperform their traditional softmax counterparts, or show comparable performance on CIFAR-10, CIFAR-100, and ImageNet-1K datasets. We also deploy Fourier KArAt to ConViT and Swin-Transformer, and use it in detection and segmentation with ViT-Det. We dissect the performance of these architectures on the classification task by analyzing their loss landscapes, weight distributions, optimizer paths, attention visualizations, and transferability to other datasets, and contrast them with vanilla ViTs. KArAt's learnable activation yields a better attention score across all ViTs, indicating improved token-to-token interactions and contributing to enhanced inference. ~~Still, its generalizability does not scale with larger ViTs.~~ However, many factors, including the present computing interface, affect the relative performance of parameter- and memory-heavy KArAts. We note that the goal of this paper is not to produce efficient attention or challenge the traditional activations; by designing KArAt, we are the first to show that attention can be learned and encourage researchers to explore KArAt in conjunction with more advanced architectures that require a careful understanding of learnable activations.

## 1 Introduction

*Artificial general intelligence* has become a rapidly growing research direction, and Kolmogorov-Arnold Network (KAN) Liu et al. (2024) marks a remarkable innovation in that. KANs with learnable activation functions can potentially capture more complex relationships and facilitate meaningful interaction between the model and human intuition. After training a KAN on a specific problem, researchers can extract the learned univariate functions that the model uses to approximate complex multivariable functions. By studying these learned functions, researchers can gain insights into the underlying relationships from the data and refine the model. KANs exhibit state-of-the-art performance in finding symbolic function representations Yu et al. (2024), continual learning of one-dimensional functions Liu et al. (2025).

KANs were integrated with neural network architectures, the primary ones being conventional MLPs or convolution neural networks (CNNs). E.g., Ferdaus et al. (2024); Bodner et al. (2024); Drokin (2024); Abueidda et al. (2025); Wang et al. (2025) combine KAN with CNNs, Li et al. (2025) combine KAN with U-Net, etc. Interestingly, (Yu et al., 2024) claimed to make the first *fairer comparison* between KANs and MLPs on multiple ML tasks on small-scale datasets. By setting the same parameter counts in both networks, Yu et al. (2024) concluded that KANs with B-Spline basis significantly underperform compared to their MLP counterparts in all ML tasks except symbolic formula representation; Azam & Akhtar (2024) reinstates a similar claim for vision. However, KAN's exploration of more advanced architectures, such as Transformers, remains limited. Chen et al. (2024) replace the MLP layers inside the encoder blocks of data-efficient image Transformers (DeiTs) Touvron et al. (2021) with KANs and proposed DeiT+KAN, Yang & Wang (2025) proposed two variants: ViT+KAN that replaces the MLP layers inside ViT's encoder blocks, and Kolmogorov-Arnold Transformer (KAT), albeit similar to ViT+KAN but with a refined group-KAN strategy.

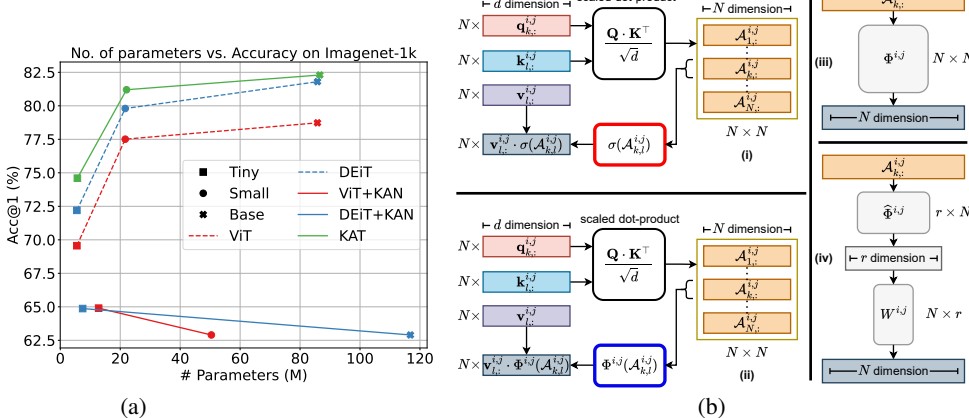

(a)  (b)

Figure 1: (a) **Model parameters vs. Top-1 accuracy** in ImegeNet-1K training of vanilla ViTs Dosovitskiy et al. (2020), Vision KAN (DeiT+KAN) by Chen et al. (2024), ViT+KAN and Kolmogorov-Arnold Transformer (KAT) by Yang & Wang (2025).(b-i) The traditional `softmax` attention. (b-ii) The **Kolmogorov-Arnold Attention (KArAt)** replaces the `softmax` with a learnable operator, $\Phi^{i,j}$. (b-iii) Regular KArAt uses an operator matrix, $\Phi^{i,j}$ with $N^2$ learnable units acting on each row of $\mathcal{A}^{i,j}$, and is prohibitively expensive. (b-iv) Modular KArAt uses an operator $\widehat{\Phi}^{i,j} \in \mathbb{R}^{N \times r}$ with $r \ll N$, followed by a learnable linear projector $W \in \mathbb{R}^{r \times N}$.

While KANs were claimed to be not superior to simple MLPs in vision tasks Yu et al. (2024); Azam & Akhtar (2024), we were curious to find out how ViTs can *see images through the lens of KANs*. In that effort, Figure 1a demonstrates the performance of vanilla ViTs (-Tiny, -Small Wu et al. (2022); Steiner et al. (2022), and -Base Dosovitskiy et al. (2020)) and their KAN counterparts on ImageNet-1K Deng et al. (2009) training. While different variants of DeiT+KAN Chen et al. (2024) and ViT+KAN Yang & Wang (2025) show a 5%–18% drop in the Top-1 accuracy compared to vanilla ViTs and DeiTs in ImageNet-1K, the KATs Yang & Wang (2025) show approximately 1%–2.5% gain in Top-1 accuracy compared to vanilla ViTs and DeiTs by keeping about the same number of parameters. These results show *KANs require further investigation in more advanced architectures such as Transformers.* Nevertheless, first, we want to understand why there is a discrepancy in these two seemingly equivalent implementations.

Searching for an answer, we realized that while DeiT+KAN and ViT+KAN replace the MLP layer with KAN in the Transformer's encoder block, KAT implements a sophisticated group-KAN strategy that reuses a learnable function among a group of units in the same layer and chooses different bases for different encoder blocks. This strategy improves accuracy by keeping almost the same parameter counts. Therefore, *simply replacing MLPs with KANs might not guarantee better performance, but a properly designed KAN could.* But, can we only *learn* attention?

Irrespective of the tasks or datasets at hand, traditional multihead attention, the heart of the Transformers, employs a softmax non-linearity as a *one-size-fits-all* mechanism for token interactions without the necessary justification for its use. We hypothesize that adopting learnable KAN operators could be a better fit for approximating the underlying non-linear relationship between image patches, providing the flexibility for modeling token interactions in ViTs. Therefore, in the rise of next-generation Transformers, such as Show-o Xie et al. (2025), RL-VLM-F Wang et al. (2024), Google's TITAN Behrouz et al. (2024), and SAKANA AI's Transformer[2] Sun et al. (2025), that mimic the human brain, we ask: *How can we deploy learnable multi-head self-attention to the (vision) Transformers?*

We note that there is a line of research that proposes efficient attention mechanisms in terms of sparse attention Rahimian et al. (2024); Yun et al. (2020); Shi et al. (2021); Kovaleva et al. (2019); Zhang et al. (2021); Zaheer et al. (2020); Guo et al. (2019) or linear/kernelized attention Han et al. (2025); Nguyen et al. (2023; 2021; 2022b); Lu et al. (2021); Nguyen et al. (2022a), to remedy computational and memory complexities of attention calculation. In contrast, this paper tries to understand how *learnable multi-head self-attention (MHSA) modules* can perform over regular self-attention used in ViTs, a defining technology in vision in the last six years. Taken together, our contributions are:

**Designing a Learnable Attention Module for ViTs.** This pilot study is on the self-attention in ViTs. In §3, we propose a general learnable Kolmogorov-Arnold multi-head self-attention, or KArAt, for ViTs that can operate on any basis. Due to the memory explosion, any full-rank learnable MHSA operator in ideal KArAt causes a training bottleneck. We propose a low-rank approximation of the operator to reduce memory requirements to a feasible level, but leave the possibility of exploring alternative techniques. By adopting the Fourier basis into this modular version, we design Fourier KArAt; see §4 for details. We cannot perform tasks with varying sequences and cross-attention

without major modifications to KArAt. The most prominent modalities other than vision, such as audio, speech, video, and text, have varying sequence length, involve modeling with cross-attention-based encoder-decoder architectures, for which token interactions would be substantially different from self-attention. These are non-trivial technical challenges for implementing KArAt.

**Benchmarking, Evaluation, and Analysis (§5-§6).** We benchmark Fourier KArAt on CIFAR-10, CIFAR-100, Krizhevsky et al. (2009) and ImageNet-1K Deng et al. (2009) datasets and evaluate their performance against vanilla ViTs (ViT-Tiny, -Small Wu et al. (2022); Steiner et al. (2022), and -Base Dosovitskiy et al. (2020)), and other popular ViTs (ConViT D'Ascoli et al. (2021) and Swin-Transformer Liu et al. (2021)). We use small-scale datasets, SVHN Netzer et al. (2011), Oxford Flowers 102 Nilsback & Zisserman (2008), and STL-10 Coates et al. (2011) for understanding the transfer learning capability of the models. Additionally, we embed KArAt with ViT-Det Li et al. (2022) for the object detection and segmentation on MS COCO Lin et al. (2014). We dissect KArAts' performance and generalization capacity by analyzing their loss landscapes, weight distributions, optimizer path, and attention visualization, and compare them with traditional softmax attention in vanilla ViTs. In addition, we investigate KArAts' explanability for visual understanding; see §5.3. We further looked into how overparameterization impacts the scalability of KArAt in larger models and showed how it can be alleviated by attention sparsification or parameter reduction; see §7.

## 2 BACKGROUND

**Multi-layer Perceptrons (MLPs)** consist of $L$ layers. By convention, $a^{[0]} = x^{[0]}$ denotes the input data and the $l^{\text{th}}$-layer is defined for input $x^{[l-1]} \in \mathbb{R}^{d_{l-1}}$ as

$$a^{[l]} = \Phi^{[l]}(W_l x^{[l-1]} + b_l), \ W_l \in \mathbb{R}^{d_l \times d_{l-1}}, b_l \in \mathbb{R}^{d_l}.$$

In MLP, the nonlinear activation functions, $\Phi(\cdot)$, are fixed Bengio et al. (2017). For supervised tasks, given a training dataset $D$ with $N$ elements of the form (input, ground-truth) pairs, $\{(x_i, y_i^\star)\}_{i=1}^N$, the loss, $\mathcal{L}(X, Y^\star | \mathcal{W}) = d_Y(a^{[L]}, Y^\star)$ (metric induced by the space $\mathbb{R}^{d_L}$) is calculated in the *forward pass*. The layerwise weights, $\{(W_l, b_l)\}_{l \in [L]}$ are learned by minimizing $\mathcal{L}(X, Y^\star | \mathcal{W})$.

**Kolmogorov-Arnold Network (KAN) Liu et al. (2025)** is a neural network involving learnable activations parametrized by a chosen set of basis functions defined on a set of grid points or knots. The idea for this network stems from the *Kolmogorov-Arnold Representation Theorem*.

**Theorem 1** (Kolmogorov-Arnold Representation Theorem). *Kolmogorov (1956) For any multivariate continuous function, $f : [0, 1]^n \to \mathbb{R}$, there exists a finite composition of continuous single-variable functions, $\phi_{q,p} : [0, 1] \to \mathbb{R}, \Phi_q : \mathbb{R} \to \mathbb{R}$ such that $f(x) = f(x_1, x_2, \cdots, x_n) = \sum_{q=1}^{2n+1} \Phi_q \left( \sum_{p=1}^n \phi_{q,p}(x_p) \right).$*

Theorem 1 describes an *exact* finite expression using 2 layers, but Liu et al. (2025) generalized this representation to multi-layers via KAN; it is analogous to the *Universal Approximation Theorem* Hornik et al. (1989). For input $x^{[0]}$, an $L$-layer KAN is given as $\text{KAN}(x^{[0]}) = \Phi^L \circ \cdots \circ \Phi^l \circ \cdots \circ \Phi^1(x^{[0]})$. In each KAN layer, the activation function $\Phi$ is learnable, and $\Phi^{[l]} = [\Phi_{ij}^{[l]}]$ operates on $x^{[l-1]} \in \mathbb{R}^{d_{l-1}}$ to produce $x^{[l]} \in \mathbb{R}^{d_l}$, such that:

$$x_i^{[l]} = \Phi_{i:}^{[l]}(x^{[l-1]}) = \sum_{j=1}^{d_{l-1}} \Phi_{ij}^{[l]}(x_j^{[l-1]}).$$

Originally, Liu et al. (2025) chose the *B-spline* basis functions. That is, the activation functions were defined to be $\phi(x) = w(b(x) + \text{spline}(x))$, where $b(x) = \text{SiLU}(x) = \frac{x}{1+e^{-x}}$ and $\text{spline}(x) = \sum_i c_i B_i(x)$, with $B_i(\cdot)$ being one of the $k$-th degree *B-spline* basis functions parametrized by the $G$ grid points on a uniform grid, $[-I, I]$. In this case, the representation for each function is given as:

$$\Phi_{ij}^{[l]}(x_j^{[l-1]}) = w_{ij}^{[l]}(\text{SiLU}(x_j^{[l-1]}) + \sum_{m=1}^{G+k-1} c_{ijm}^{[l]} B_{ijm}(x_j^{[l-1]})).$$

The weights $\{([c_{ijm}^{[l]}], [w_{ij}^{[l]}])\}_{l=1}^L$ are learned by minimizing the loss for a supervised learning task. The bases for KANs' activations could be Fourier Mehrabian et al. (2024); Xu et al. (2024), wavelet Bozorgasl & Chen (2024), fractals Yang & Wang (2025), etc.

**Multi-head self-attention (MHSA) in ViTs Dosovitskiy et al. (2020).** The encoder-only vanilla ViT architecture is inspired by the Transformer proposed in Vaswani et al. (2017). For simplicity, we do not mention the details of the layer normalization and other technicalities. Our central focus is the MHSA architecture. For limited space, we move this discussion to §A.

## 3    How Can We Design Learnable Attention?

Last year, we witnessed a surge in embedding KANs in different DNN architectures Genet & Inzirillo (2024); Ferdaus et al. (2024); Li et al. (2025). To our knowledge, the work closely related to ours is KAT, where Yang & Wang (2025) replaces the MLP layers in the ViTs with KAN layers; also, see Chen et al. (2024). We note that, for attentive graph neural networks (GNNs), Fang et al. (2025) unifies the scoring functions of attentive GNNs and names it as Kolmogorov-Arnold Attention (KAA). However, it is orthogonal to our MHSA design; deploying learnable MHSA is complicated. In this Section, we design *learnable multi-head self-attention (MHSA) module* for ViTs, with improved interpretability, adaptability, and expressiveness that operate inside its encoder blocks.

Let $\mathcal{A}^{i,j} \in \mathbb{R}^{N \times N}$ be the attention matrix for $i^{\text{th}}$ head in the $j^{\text{th}}$ encoder block. Instead of using the softmax function row-wise, we can use a learnable activation function, $\tilde{\sigma}$ on the row vectors of each attention head $\mathcal{A}^{i,j}$. With any choice of the basis functions (e.g., B-Spline, Fourier, Wavelets, etc.), the activated attention row vector, $\tilde{\sigma}(\mathcal{A}_{k,:}^{i,j})$ for $k \in [N]$ can be written as

$$\tilde{\sigma}\left[(\mathcal{A}_{k,:}^{i,j})\right] = \begin{pmatrix} \phi_{11}(\cdot) & \phi_{12}(\cdot) & ... & \phi_{1N}(\cdot) \\ \phi_{21}(\cdot) & \phi_{22}(\cdot) & ... & \phi_{2N}(\cdot) \\ \vdots & \vdots & \ddots & \vdots \\ \phi_{N1}(\cdot) & \phi_{N2}(\cdot) & ... & \phi_{NN}(\cdot) \end{pmatrix} \begin{pmatrix} \mathcal{A}_{k,1}^{i,j} \\ \mathcal{A}_{k,2}^{i,j} \\ \vdots \\ \mathcal{A}_{k,N}^{i,j} \end{pmatrix} = \left(\Phi^{i,j}\left[(\mathcal{A}_{k,:}^{i,j})^{\top}\right]\right)^{\top}, \tag{1}$$

where $\Phi^{i,j} = [\phi_{pq}^{i,j}] \in \mathbb{R}^{N \times N}$ and each matrix entry, $\phi_{pq}$, is referred to as a *learnable unit*, represented using a set of basis functions $\{\psi_m^{i,j}\}_{m=1}$. The coefficients associated with the basis functions are the *learnable parameters*. In our convention, $\mathcal{A}_{k,:}^{i,j} \in \mathbb{R}^{1 \times N}$ is a row vector. We apply the transpose operation to each row vector to adopt the convention used in KAN layers. Hence $\Phi^{i,j}\left[(\mathcal{A}_{k,:}^{i,j})^{\top}\right] \in \mathbb{R}^{N \times 1}$, and we transpose it to obtain learnable attention row vectors $\tilde{\sigma}\left[(\mathcal{A}_{k,:}^{i,j})\right]$; see Figures 1b(ii)-(iii).

**Projection onto the probability simplex.** The softmax function acts on each row of $\mathcal{A}$ to create a probability distribution; the learnable attention does not guarantee that. To ensure that each row vector of $\Phi(\mathcal{A})$ lies on a probability simplex, we project them onto the $\ell_1$-unit ball. That is, for each attention matrix, $\mathcal{A}^{i,j} \in \mathbb{R}^{N \times N}$, and for each $k \in [N]$, we want to have $\|\tilde{\sigma}\left[(\mathcal{A}_{k,:}^{i,j})\right]\|_1 = 1$ and $\tilde{\sigma}\left[(\mathcal{A}_{k,l}^{i,j})\right] \geq 0$ for $l \in [N]$. We cast a sparse approximation problem, whose variants have been well-studied in the past decade and arise frequently in signal processing Bryan & Leise (2013); Donoho (2006); Dutta (2016) and matrix approximation Boas et al. (2017); Stewart (1993); see §3. Algorithm 1 provides an $\ell_1$-projection pseudocode; for each $k \in [N]$, setting $y = \tilde{\sigma}\left[(\mathcal{A}_{k,:}^{i,j})\right]$ and $m = N$ in Algorithm 1, we obtain the projected attention vector.

## 4    Our Architecture

Different from learning activations for MLP, as in KAN and KAT, implementing the generic learnable attention incurs impractical computational and memory costs on present computing hardware and DL toolkits. For learnable activation, $\tilde{\sigma} : \mathbb{R}^N \to \mathbb{R}^N$, a full-rank operator, $\Phi^{i,j} \in \mathbb{R}^{N \times N}$ acts on each row $\mathcal{A}_{k,:}^{i,j}$ of $\mathcal{A}^{i,j}$, and produces $\tilde{\sigma}(\mathcal{A}_{k,:}^{i,j})$. These operations are extremely compute-heavy and have large memory footprints. E.g., ViT-Tiny Dosovitskiy et al. (2020); Wu et al. (2022); Steiner et al. (2022) has 5.53M parameters and requires nearly 0.9 GB of GPU memory to train. Implementing the learnable attention equation 1 in ViT-Tiny training with B-splines of order 5 and grid size 10 and Fourier basis with grid size 10 increases the parameter count to 30.68M and 39.06M, respectively. We could not train the version with B-splines for its humongous memory requirements, and B-spline computations are non-parallelizable. The one with the Fourier basis is parallelizable, but it takes approximately 60 GB of GPU memory when computing with a batch of one image. This computational bottleneck is agnostic of the basis functions.

**How can we remedy this?** Deep neural networks exhibit low-rank structure Oja (1982); Jain et al. (2016). Recently, Kwon et al. (2024) showed that across various networks, the training largely occurs

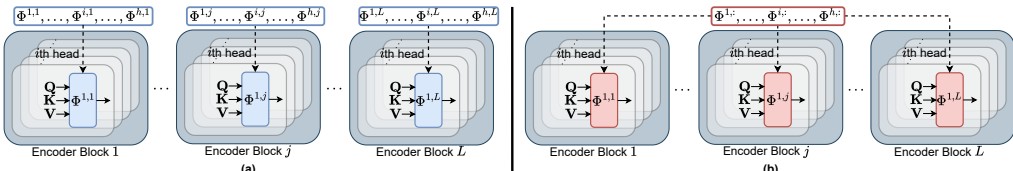

Figure 2: **Different configurations to update** $\widehat{\Phi}$: (a) Blockwise configuration, where $\Phi^{i,1} \neq \Phi^{i,2} \neq \cdots \neq \Phi^{i,L}$ for all $i = 1, 2, ..., h$; (b) universal configuration, where $\Phi^{i,1} = \Phi^{i,2} = \cdots = \Phi^{i,L} = \Phi^i$ for all $i = 1, 2, ..., h$.

within a low-dimensional subspace. Therefore, we postulate that the attention matrices also have an underlying low-rank structure.

To validate this, we perform spectral analysis of the attention matrices before and after the `softmax` activation; see Figures 5a-5b. The *scree test* Cattell (1966) shows that for traditional attention matrices without softmax, there are 8 significant singular values; after softmax activation, this count increases to 16. The above observation verifies that with or without the nonlinear activation, the attention matrix $\mathcal{A}^{i,j}$ possesses an underlying low-rank structure. This motivates us to use a lower-dimensional operator, $\widehat{\Phi}$ for the learnable attention calculation with $r = 12$. In §C.4.2-C.4.3, we perform an ablation to find the best $r$.

Instead of $\Phi^{i,j} \in \mathbb{R}^{N \times N}$, we use an operator, $\widehat{\Phi}^{i,j} \in \mathbb{R}^{r \times N}$ such that $r \ll N$, and the new learned activation is $r$-dimensional for $k \in [N]$. This process significantly reduces the computational overhead. Next, we post-multiply another learnable weight matrix, $W^{i,j} \in \mathbb{R}^{N \times r}$ to project them back to their original dimension. For each $k \in [N]$, this operation results in computing $\widehat{\sigma}(\mathcal{A}_{k,:}^{i,j}) = \left[ W^{i,j} \widehat{\Phi}^{i,j} \left[ (\mathcal{A}_{k,:}^{i,j})^{\top} \right] \right]^{\top}$ (Figure 1b(iv)); see other variants in §C.5. Our approximation is not unique or the best; it is one of the feasible solutions given the present computing interface.

**Fourier Kolmogorov-Arnold Attention.** Although the default basis for KANs is B-Splines in Liu et al. (2025), the number of MHSA parameters is much more than that of an MLP. Specifically, for $L$ encoder blocks, each with $h$ attention heads, the parameter complexity is $O(N^2 Lh(G + k))$, for $\Phi \in \mathbb{R}^{N \times N}$ and $O(NrLh(G + k))$, for $\widehat{\Phi} \in \mathbb{R}^{r \times N}$, if the model uses B-Splines of degree $k$.

Moreover, as Yang & Wang (2025) mentioned, B-splines are localized functions, not standard CUDA functions. Although efficient CUDA implementations for cubic B-Splines exist Ruijters & Thévenaz (2012); Ruijters et al. (2008), their overall implementation results in slower, sparse, non-scalable, and complicated GPU execution; also, see Xu et al. (2024); Pal & Das (2024). In §C.4.1, our extensive experiments with B-spline basis in KAN for image classification tasks verify their poor generalizability on medium-scale datasets (e.g., CIFAR-10 and CIFAR-100).

So, what could be an attractive basis? A fundamental question in function approximation is whether the function converges *pointwise almost everywhere*. Carleson in 1966 proved the following fundamental result for the Fourier approximation of $L^p$ periodic functions.

**Theorem 2.** *Carleson (1967) Let $f$ be an $L^p$ periodic function for $p \in (1, \infty]$, with Fourier coefficients $\hat{f}(n)$. Then $\lim_{N \to \infty} \sum_{|n| \leq N} \hat{f}(n)e^{inx} = f(x)$, for almost every $x$.*

In particular, if a function is continuously differentiable, its Fourier series converges to it everywhere. In the past, Xu et al. (2024); Mehrabian et al. (2024) used the Fourier basis in KAN, Dong et al. (2024) proposed a Fourier analysis network (FAN) to capture periodic phenomena. Motivated by these, we use the Fourier basis to approximate the effect of the smooth softmax function. We employ the Fourier basis, $\{\sin(\cdot), \cos(\cdot)\}$ with gridsize $G$ to design learnable attention. Algorithm 1 can be used optionally. For $\widehat{\Phi}^{i,j} = [\widehat{\phi}_{pq}^{i,j}] \in \mathbb{R}^{r \times N}$, we have $\widehat{\Phi}^{i,j} : \mathbb{R}^N \to \mathbb{R}^r$. Hence, each row $\mathcal{A}_{k,:}^{i,j}$, for $k \in [N]$, transformed into $\widehat{\Phi}_p^{i,j} \left[ (\mathcal{A}_{k,:}^{i,j})^{\top} \right] = \sum_{q=1}^{N} \widehat{\phi}_{pq}^{i,j}(\mathcal{A}_{k,q}^{i,j})$, via the Fourier bases such that

$$\widehat{\phi}_{pq}^{i,j}(\mathcal{A}_{k,q}^{i,j}) = \sum_{m=1}^{G} a_{pqm} \cos(m\mathcal{A}_{k,q}^{i,j}) + b_{pqm} \sin(m\mathcal{A}_{k,q}^{i,j}), \tag{2}$$

where $m$ refers to the $m^{\text{th}}$ grid point on a uniform grid of size $G$. The weights, $[\{[a_{pqm}], [b_{pqm}]\}_{m=1}^{G}, W^{i,j}]$ are updated in the backpropagation. If we use a Fourier basis in $\widehat{\Phi} \in \mathbb{R}^{r \times N}$, the parameter complexity of MHSA becomes $O(2NrhGL)$. Regardless of the basis used, the linear projection takes $O(N^2 rhL)$ FLOPs. For any Transformer, $L$ and $h$ are fixed. So, one may omit them from the complexity results. See FLOPs and memory requirement in Table 15.

**Blockwise and Universal operator configuration.** We consider two configurations for updating the operator $\widehat{\Phi}^{i,j}$. (*a*) **Blockwise**: In this configuration, each encoder block learns the attention

Table 1: Performance of the best-performing Fourier KArAt models compared to the conventional vanilla ViT baselines. The best and the second-best Top-1 accuracies are given in red and blue, respectively. The ↓ and ↑ arrows indicate the relative loss and gain, respectively, compared to the base models.

| Model | CIFAR-10 | | CIFAR-100 | | ImageNet-1K | | Parameters |
|---|---|---|---|---|---|---|---|
| | Acc.@1 | Acc.@5 | Acc.@1 | Acc.@5 | Acc.@1 | Acc.@5 | |
| ViT-Base | 83.45 | 99.19 | 58.07 | 83.70 | 72.90 | 90.56 | 85.81M |
| $+ G_1B$ | 81.81 (1.97%↓) | 99.01 | 55.92(3.70%↓) | 82.04 | 68.03(6.68%↓) | 86.41 | 87.51M (1.98% ↑) |
| $+ G_1U$ | 80.75(3.24%↓) | 98.76 | 57.36 (1.22% ↓) | 82.89 | 68.83 (5.58%↓) | 87.69 | 85.95M (0.16% ↑) |
| ViT-Small | 81.08 | 99.02 | 53.47 | 82.52 | 70.50 | 89.34 | 22.05M |
| $+ G_3B$ | 79.78 (1.60%↓) | 98.70 | 54.11 (1.20%↑) | 81.02 | 67.77 (3.87%↓) | 87.51 | 23.58M (6.94%↑) |
| $+ G_3U$ | 79.52(1.92%↓) | 98.85 | 53.86(0.73%↑) | 81.45 | 67.76(3.89%↓) | 87.60 | 22.18M (0.56%↑) |
| ViT-Tiny | 72.76 | 98.14 | 43.53 | 75.00 | 59.15 | 82.07 | 5.53M |
| $+ G_3B$ | 76.69(5.40%↑) | 98.57 | 46.29(6.34%↑) | 77.02 | 59.11 (0.07%↓) | 82.01 | 6.29M (13.74%↑) |
| $+ G_3U$ | 75.56 (3.85%↑) | 98.48 | 46.75(7.40%↑) | 76.81 | 57.97(1.99%↓) | 81.03 | 5.59M (1.08%↑) |

Table 2: Performance of Fourier KArAt fine-tuned on small datasets from ImageNet-1K pre-trained weights.

| Model | Acc.@1 | | |
|---|---|---|---|
| | SVHN | Flowers 102 | STL-10 |
| ViT-Base | 97.74 | 92.24 | 97.26 |
| $+ G_1B$ | 96.83 | 89.66 | 95.30 |
| $+ G_1U$ | 97.21 | 89.43 | 95.78 |
| ViT-Small | 97.48 | 91.46 | 96.09 |
| $+ G_3B$ | 97.04 | 89.67 | 95.26 |
| $+ G_3U$ | 97.11 | 90.08 | 95.45 |
| ViT-Tiny | 96.69 | 84.21 | 93.20 |
| $+ G_3B$ | 96.37 | 83.67 | 93.09 |
| $+ G_3U$ | 96.39 | 83.70 | 92.93 |

Table 3: Performance comparison of Fourier-KArAt and `softmax` attention on various vision transformer architectures.

| Model | CIFAR-10 | | ImageNet-1K | |
|---|---|---|---|---|
| | Acc.@1 | Acc.@5 | Acc.@1 | Acc.@5 |
| ConViT-Tiny | 71.36 | 97.86 | 57.91 | 81.79 |
| $+ G_3B$ | 75.57 | 98.61 | 56.57 | 80.75 |
| $+ G_3U$ | 74.51 | 98.63 | 56.51 | 80.93 |
| Swin-Tiny | 84.83 | 99.43 | 76.14 | 92.81 |
| $+ G_3B$ | 79.34 | 98.81 | 73.19 | 90.97 |

Table 4: **Attention Transfer** from ViT+Fourier-KArAt to vanilla ViTs, following Li et al. (2024).

| Model | Attention | CIFAR-10 | | CIFAR-100 | |
|---|---|---|---|---|---|
| | Transfered From | Acc.@1 | Acc.@5 | Acc.@1 | Acc.@5 |
| ViT-Tiny | None | 72.76 | 98.14 | 43.53 | 75.00 |
| ViT-Tiny+$G_3B$ | None | 76.69 | 98.57 | 46.29 | 77.02 |
| ViT-Tiny | ViT-Tiny+$G_3B$ | 77.94 | 98.71 | 48.91 | 78.42 |
| ViT-Base | None | 83.45 | 99.19 | 58.07 | 83.70 |
| ViT-Base+$G_1B$ | None | 81.81 | 99.01 | 55.92 | 82.04 |
| ViT-Base | ViT-Base+$G_1B$ | 81.34 | 98.70 | 55.33 | 80.80 |
| ViT-Base | ViT-Tiny+$G_3B$ | 80.39 | 99.02 | 56.11 | 82.44 |

through $h$ distinct operators $\widehat{\Phi}$ for each of the $h$ heads, totaling $hL$ operators; see Figure 2 (a). Like the MHSA in vanilla ViTs, the blockwise configuration is designed to learn as many different data representations as possible. (b) **Universal**: This is motivated from the KAT Yang & Wang (2025). In KAT, the MLP head is replaced with different variations of a KAN head—KAN and Group-KAN. In Group-KAN, the KAN layer shares rational base functions and their coefficients among the edges. Inspired by this, in our update configuration, all $L$ encoder blocks share the same $h$ operators; $\widehat{\Phi}^{i,j} = \widehat{\Phi}^i$ for $j = 1, 2, \ldots, L$; see Figure 2-(b). Rather than learning attention through $hL$ operators, this configuration only uses $h$ operators. Here, we share all learnable units and their parameters across $L$ blocks in each head. We postulate that the blockwise mode with more operators captures more nuances from the data. In contrast, the universal mode is suitable for learning simpler decision boundaries from the data. We also note that these two modes can be used with equation 1 as shown in Figure 1b. Finally, Algorithm 2 in the Appendix provides the pseudocode for applying Fourier-KARAT in the ViT encoder block; see more nuanced discussion about them in §C.4.6.

## 5 EMPIRICAL STUDY

**Implementing KArAt.** We provide the implementation details in §C.1; see §C.1.1 for baselines and datasets. We integrate 5 wavelet bases, rational basis, Fourier basis, and 3 different base activation functions in our learnable MHSA design; see §C.1.3. Fourier KArAt performs the best, and we report it in the main paper. It has 1 hyperparameter, the grid size, $G$, and 2 configurations, blockwise or universal. We vary grid sizes from the set $G = \{1, 3, 5, 7, 9, 11\}$. For each value, we perform universal and blockwise updates; see Figure 2. With this formalization, $G_nB$ and $G_nU$ denote Fourier-KArAt with grid size $n$ and weights updated in blockwise and universal mode, respectively. See the ablation study of these variants in §C.5. We dispense the $\ell_1$ projection, as projecting the learned attention vectors to a probability simplex degrades the performance; see §C.4.5.

### 5.1 TRAINED MODEL QUALITY, TRANSFERABILITY, AND GENERALIZABILITY

(*i*) **Image Classification.** Table 1 presents the performance of the best-performing Fourier KArAt variants; see Figure 6 for the training loss and test accuracy curves. Fourier KArAt is easily optimized in the initial training phase, gaining at par or better loss value and accuracy than the vanilla ViTs. However, in the later training phase, except for the fewer parameter variants (ViT-Tiny+Fourier KArAt), the loss curve flattens faster; we infer that for most of the Fourier-KArAt scenarios. We also notice that larger models like ViT-Base are more susceptible to slight changes in the parameters, making the training process challenging to manage, which we try to investigate in the rest of the paper using several analytical studies. Overall, ViT-Tiny+Fourier KArAt variants outperform ViT-Tiny on CIFAR-10 and CIFAR-100 by $5.40\%$ and $7.40\%$, respectively, and on ImageNet-1K, it achieves a similar accuracy. ViT-Small and -Base models with Fourier KArAt variants can barely outperform the vanilla models, and the accuracy differences increase as we tackle larger base models.

(*ii*) **Performance on other ViTs for Image Classification.** We fused KArAt with modern ViT-based architectures, ConViT D'Ascoli et al. (2021) and Swin Transformer Liu et al. (2021); §C.1.4 provides

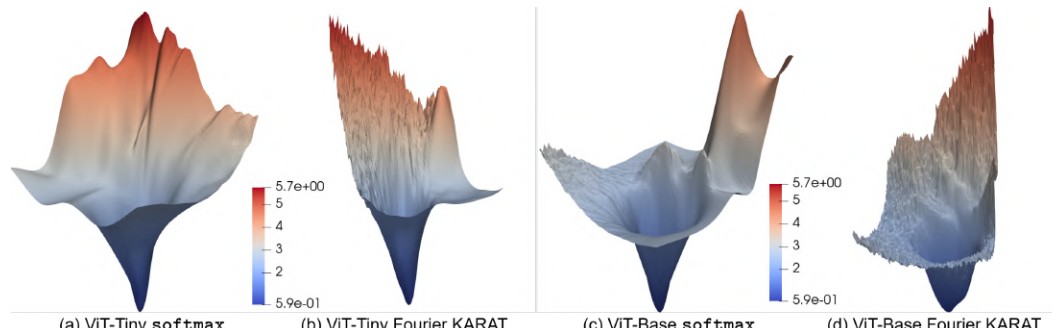

(a) ViT-Tiny `softmax`  (b) ViT-Tiny Fourier KARAT  (c) ViT-Base `softmax`  (d) ViT-Base Fourier KARAT

Figure 3: **3D-visualization of Loss landscape for ViT-Tiny and ViT-Base** along the two largest principal component directions of the successive change of model parameters. KArAt's loss landscapes are significantly less smooth than those of traditional attention; spiky loss landscapes are undesirable for optimization stability and the generalizability of the resulting model. See **Figure 10** for the **loss contours and the optimizer trajectory.**

their implementation details. Table 3 shows that Fourier KArAt outperforms ConViT on CIFAR-10, experiences a modest drop on ImageNet-1K, and shows lower performance for hierarchical models, such as Swin, which requires a careful investigation.

(*iii*) *Transferability*: **Image Classification.** Unlike ViTs, which were pre-trained on large datasets and then transferred to various mid-sized or small image recognition benchmarks Dosovitskiy et al. (2020), which helped them achieve state-of-the-art results, KArAts were never pre-trained with such datasets to showcase their full potential. Due to limited computing resources, we could not perform large-scale training. We investigate KArAts' transfer capability by fine-tuning on smaller datasets like SVHN Netzer et al. (2011), Oxford Flowers 102 Nilsback & Zisserman (2008), and STL-10 Coates et al. (2011) from their ImageNet-1K pre-trained weights; see dataset details in §C.1.1. Table 2 shows KArAts transfer well across all datasets and have comparable performance with vanilla ViTs, even when their ImageNet-1K performance was not equivalent.

(*iv*) *Generalizability*: **Detection & Segmentation.** To understand the robustness of the Fourier KArAt, we employ ViT backbones for detection and segmentation; see results and discussion in §C.2.

**Key Takeaways.** (*a*) The Top-1 accuracies from Tables 1 and 2 show that Blockwise configuration is more desirable over Universal for image classification. (*b*) From Table 2, we can infer that the Fourier KArAt transfers well in fine-tuning tasks. However, the transferability of hyperparameters, e.g., grid size $G$, across datasets remains an open question. (*c*) KArAt generalizes well. For random initialization in detection, the vanilla ViT-Det has an advantage over KArAt, and thus, we see a small performance gap. The best KArAt hyperparameters for this task are yet to be found. Overall, KArAt shows significant potential if the incompatibilities are properly addressed.

## 5.2 PERFORMANCE ANALYSIS

We dissect the performance of Fourier KArAt using the following tools:

(*i*) **Loss Landscape.** We visualize the loss landscape of KArAts to understand why they converge slowly in the later training phase, and why their generalizability does not scale with larger models. Following Li et al. (2018), we plot the loss surface along the directions in which the gradients converge; see §C.1.6 for implementation. Figure 3 shows that Fourier KArAt in ViT architectures significantly impacts the smoothness of the loss surfaces. ViT-Tiny, with the fewest parameters, has the smoothest loss landscape and modest generalizability. In contrast, ViT-Tiny+Fourier KArAt's loss landscape is spiky; it indicates the model is full of *small-volume minima* Huang et al. (2020). However, the model is modest in the number of parameters, so the gradient descent optimizer can still find an optimized model with better generalizability than the vanilla ViT-Tiny; hence, it gains a better test accuracy; see Table 1. ViT-Base, however, has more parameters than Tiny, and its loss surface is much spikier than ViT-Tiny. Finally, the loss surface of ViT-Base+Fourier KArAt is most spiky, making it a *narrow margin model with sharp minima* in which small perturbations in the parameter space lead to high misclassification due to their exponentially larger volume in high-dimensional spaces; see Figure 3(d). Moreover, ViT-Base+$G_3B$ has 14 times more parameters than ViT-Tiny+$G_1B$. With learnable activations, gradient descent optimizers fail to find the best-optimized model, as small differences in the local minima translate to exponentially large disparities. The increasing number of local minima in the later phases of the training causes the slow convergence.

(*ii*) **Attention Visualization.** MHSA in vanilla ViTs captures region-to-region interaction in images. DINO Caron et al. (2021) provides an innovative way to explain the dominant regions in an image that

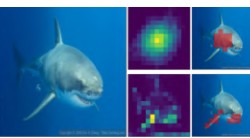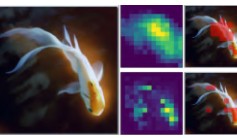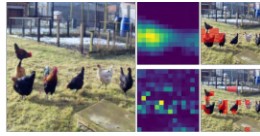

Figure 4: **Vit-Tiny Attention map visualization.** Original images for inference (the left), the attention score (middle), and image regions of the dominant head (Top row: Fourier KArAt, bottom row: traditional MHSA).

contribute towards the inference decision. It maps the self-attention values of the `CLS` token at the last encoder layer, which is directly responsible for capturing the information related to the class label. While the computer vision community predominantly uses this attention visualization, it is unsuitable for our case. Unlike `softmax`, Fourier KArAt does not inherently ensure learned attention values in $[0, 1]$, as we dispense the $\ell_1$ projection. Therefore, we cannot interpret its performance similarly. Nonetheless, pre-softmax or pre-KArAt values in the attention matrix $\mathcal{A}^{i,j}$ are supposed to capture token-to-token interactions, and we adapt the attention maps to ignore the negative values for the sake of visualization. We use the ImageNet-1K trained models of ViT-Tiny (with traditional MHSA and Fourier KArAt) and plot the attention maps of the last layers in Figures 4 and 11.

### 5.3 KArAt's Flexibility over Traditional Attention in ViT

Stemming from Kolmogorov-Arnold Representation Theorem (Theorem 1) Kolmogorov (1956), KANs Liu et al. (2025) with learnable activation functions can approximate complex mathematical functions and symbolic representations, and provide an interpretable alternative to MLPs Liu et al. (2025; 2024); Yang & Wang (2025). Precisely, if the target function admits a structure of the composition of $L$-KAN layers with smooth activation functions, then one can use an $L$-layer KAN with B-Spline basis to approximate it well [Theorem 2.1 in Liu et al. (2025)]. This allows them to model complex relations that lie in the data distribution; KANs can discover the need for a new function whose numerical behavior suggests it may be a Bessel function (Figure 23(d) in Liu et al. (2025)), and the authors also show that KANs can discover unknown equations (Figure 4(e)).

By using a similar argument, we hypothesize that KANs could be a better fit for approximating the underlying non-linear relationship between image patches and can be used for modeling token interactions with learnable components. Hence, KArAt can be viewed as an upgrade to vanilla self-attention, offering adaptability and more freedom for modeling token interactions in an interpretable way. KArAt provides the freedom to choose from a multitude of simple, easy-to-interpret, and tunable functions whose linear combination can produce a large class of unknown functions, rather than a fixed and known function, `softmax`. The traditional `softmax` attention adds a non-linearity for modeling such interactions in a *one-size-fits-all* mechanism without any justification. This interpretability is directly reflected in KArAt's attention maps; see Figures 4 and 11.

**Why KArAt's attention maps are more explainable?** DINO Caron et al. (2021), being self-supervised, attends to the entire object and obtains richer features and deep visual representations Ericsson et al. (2021). To attend to the entire object like self-supervision, Chefer et al. (2022) in their supervised training, optimizes the attention map with additional loss objectives, which improves model interpretability and explanability. KArAt's attention maps in Figures 4 and 11 capture entire objects without any extra loss, unlike some contours in the traditional `softmax` attention. It provides improved explainability and visual understanding, compared to spurious cues in supervised ViTs.

To show that KArAt's attention maps are not just visually better, but the learnable attention leads to effective token interactions modeling, we follow the study by Li et al. (2024). To understand how the attention modulates the performance of ViTs, Li et al. (2024) considered attention matrices from a masked auto-encoder He et al. (2022), trained with self-supervision, having superior performance, and used them in a vanilla ViT, ignoring the query and key, and training the rest of the network. This study shows that attention is the most important component in a transformer and can bring improved performance over the vanilla supervised ViT. In the same spirit, we use the precomputed attention matrices from the KArAt models to train the value branch of a vanilla ViT from scratch; key and query branches remain frozen. See the implementation details in §C.1 and the results in Table 4.

ViT-Tiny, trained with attention maps of ViT-Tiny+$G_3B$, outperforms the vanilla ViT-Tiny and surpasses the performance of the ViT-Tiny+$G_3B$ for CIFAR-10 and CIFAR-100 datasets. This aligns with Li et al. (2024) as the superior attention maps of KArAt improve the performance, and the model outperforms the KArAt variants as it does not suffer from overparameterization and a spiky loss landscape; see Figure 3. In contrast, ViT-Base+$G_1B$ shows inferior performance to vanilla ViT-Base due to overparameterization and a spiky loss landscape, and hence produces a subpar model with poor attention maps. Thus, the ViT-Base trained with attention matrices from ViT-Base+$G_1B$ fails to surpass the performance of the vanilla ViT-Base or its KArAt variant. For vanilla ViT-Base, trained

with the superior attention maps of ViT-Tiny+$G_3B$, it could not improve the performance due to the redundancy of attention maps and the lack of attention head diversity Chen et al. (2022).

## 6 DISCUSSION

The learnable MHSA design raises many interesting questions regarding KArAt's computing scalability and the resulting models' generalizability. Below, we discuss their *potential positive (+) and negative (−)* implications and encourage researchers to elaborate on them.

**Designing KArAt self-attention is *not* parameter efficient and they are memory hungry (−).** Having learnable activations in the MHSA causes a memory explosion on the present GPUs. Hence, calculations with the full-rank operator, $\Phi \in \mathbb{R}^{N \times N}$, are prohibitively expensive, regardless of the basis. Present GPUs and CUDA functions are not optimized for KAN implementation as noted by many before us; see Yang & Wang (2025); Ruijters & Thévenaz (2012); Ruijters et al. (2008); Xu et al. (2024); Pal & Das (2024). Due to this, although parameter increase is not always significant (Table 1), optimized Fourier KArAt variants utilize $2.5 − 3\times$ more GPU memory compared to traditional `softmax` attention. We propose a low-rank approximation to reduce memory requirements to a feasible level (Table 15), but researchers should explore alternative techniques. Interestingly, while there is a significant training time discrepancy between vanilla ViTs and Fourier-KArAts, their inference speeds are comparable on the present compute interface and hardware configuration; see Figure 7d in §C.3. Hence, we hope the future holds more efficient system-level optimization recipes for training KArAt. Additionally, we witnessed *KArAts' inconsistent training behavior with parameter changes* in §C.4.4. Each ViT model with particular Fourier-KArAt variants has a typical $G$ value that brings out its best performance; there is no universal $G$ to follow.

**KArAts show a better attention score (+) and transfer learning capability (+) for all ViTs, but their generalizability does not scale with larger ViTs (−).** Fourier KArAt concentrates high interaction scores on the entire object compared to spurious cues Chefer et al. (2022) like vanilla ViTs Dosovitskiy et al. (2020); Touvron et al. (2021). They exhibit decent transfer learning and generalizability. The distribution of weights guarantees the stability of the training. Hence, by Zhang et al. (2022), we can postulate that first-order optimization algorithms (e.g., ADAM) can optimize KArAt's loss landscape. But Zhang et al. (2022) cannot explain why KArAt's generalizability does not scale with larger models due to high parameterization. We encourage researchers to investigate these models' local or global optimality and generalizability through theoretical and empirical studies. Moreover, we examined the scalability problem of KArAt in larger overparameterized models and demonstrated that this generalizability can be improved by investigating further how to reduce parameters or make the training more tractable and stable; see §7.

**Overparameterized KArAts' weights lie in a much lower-dimensional subspace (+).** Spiky loss landscape with local minima is probably the major impediment towards KArAt's scalability; see Figure 3. Modern transformers are highly over-parameterized, and their weight matrices show low-rank structures Kwon et al. (2024); spectral analysis (§C.7) of KArAt shows its attention matrices possess *a better low-rankness than traditional attention* (Figure 13). Considering this, properly guided parameter reduction in KArAt can result in an optimized model that generalizes well at scale. We introduced two such potential directions in §7:*(i) selective use of KArAt*, and *(ii) attention sparsification*, and showed that KArAt can be scalable when used with such properly guided strategies.

**KArAts can provide better interpretability for smaller models (+).** KANs are claimed to be more interpretable and accurate than MLPs due to their flexible choice of univariate functions Liu et al. (2025). It is evident from KArAts' attention maps, which are directly responsible for performance improvement; see §5.3. ViT-Tiny+KArAts outperforms vanilla ViT-Tiny on small-scale datasets, and performs at par on ImageNet-1K (Table 1) and transfer learning (Table 2). In the future, KArAt's learnable activation makes a case for investigating it to add more explainability in vision tasks and can find more interpretability in smaller models trained in a limited data scenario.

## 7 IMPROVING KARAT'S SCALABILITY

The fundamental challenge of training KArAt via the modern deep learning toolkits in the present-day GPU and CUDA interface remains its scalability with larger overparameterized models. Although the system-level modification is beyond the scope of this paper, in this section, we discuss two potential solutions that can alleviate *the curse of overparametrization* in KArAt:

(*i*) **Modeling Token Interactions in Chosen Layers.** ViTs learn coarse features in the initial layers, and more refined features towards the final layers; the final layers of ViTs also show higher attention entropy Hyeon-Woo et al. (2023). Therefore, self-attention plays a crucial role in the final model decision and prediction Park & Kim (2022), and ViTs attend to more tokens in the final layers. We

Table 5: Performance of the ViT variants with Fourier KArAt in selected layers compared to the conventional vanilla ViT baselines. The ↓ and ↑ arrows indicate the relative loss and gain, respectively, compared to the base models.

| Model | CIFAR-10 | | CIFAR-100 | | ImageNet-1K | | Parameters |
|---|---|---|---|---|---|---|---|
| | Acc.@1 | Acc.@5 | Acc.@1 | Acc.@5 | Acc.@1 | Acc.@5 | |
| ViT-Base | 83.45 | 99.19 | 58.07 | 83.70 | 72.90 | 90.56 | 85.81M |
| + $[G_3B]_{12}$ | 86.74 (3.94%↑) | 99.51 | 64.07 (10.33%↑) | 87.29 | 72.33 (0.78%↓) | 89.99 | 86.82M (1.18%↑) |
| + $G_3B^\dagger$ | 86.92 (4.16%↑) | 99.42 | 63.49 (9.33%↑) | 88.14 | - | - | 88.87M (3.57%↑) |
| + $G_3U^\dagger$ | 84.82 (1.64%↑) | 99.24 | 64.17 (10.50%↑) | 87.93 | - | - | 86.06M (0.29%↑) |
| ViT-Small | 81.08 | 99.02 | 53.47 | 82.52 | 70.50 | 89.34 | 22.05M |
| + $[G_3B]_{12}$ | 83.86 (3.43%↑) | 99.39 | 60.16 (12.51%↑) | 86.44 | 69.96 (0.77%↓) | 88.89 | 22.18M (0.56%↑) |
| + $G_3B^\dagger$ | 84.23 (3.89%↑) | 99.30 | 59.27 (10.85%↑) | 85.87 | - | - | 23.58M (6.94%↑) |
| + $G_3U^\dagger$ | 84.06 (3.68%↑) | 99.19 | 60.18 (12.55%↑) | 86.30 | - | - | 22.18M (0.56%↑) |
| ViT-Tiny | 72.76 | 98.14 | 43.53 | 75.00 | 59.15 | 82.07 | 5.53M |
| + $[G_3B]_{12}$ | 77.93 (7.11%↑) | 98.76 | 50.54 (16.10%↑) | 80.54 | 58.98 (0.29%↓) | 81.91 | 5.59M (1.08%↑) |
| + $G_3B^\dagger$ | 77.81 (6.94%↑) | 98.76 | 48.33 (11.03%↑) | 78.68 | - | - | 6.29M (13.74%↑) |
| + $G_3U^\dagger$ | 79.08 (8.69%↑) | 98.91 | 49.33 (13.32%↑) | 79.40 | - | - | 5.59M (1.08%↑) |

Table 6: Performance of ViT variants with Fourier KArAt in selected layers, fine-tuned on small datasets from ImageNet-1K pre-trained weights, compared to vanilla counterparts with traditional attention.

| Model | Acc.@1 | | |
|---|---|---|---|
| | SVHN | Flowers 102 | STL-10 |
| ViT-Base | 97.74 | 92.24 | 97.26 |
| + $[G_3B]_{12}$ | 97.81 | 91.97 | 96.65 |
| ViT-Small | 97.48 | 91.46 | 96.09 |
| + $[G_3B]_{12}$ | 97.56 | 91.59 | 95.79 |
| ViT-Tiny | 96.69 | 84.21 | 93.20 |
| + $[G_3B]_{12}$ | 96.40 | 85.53 | 93.35 |

hypothesize that such layers involve more complex relationships between token interactions than the initial layers. To this end, we use the learnable Fourier KArAt for token interaction modeling for the later layers and use traditional `softmax` attention in the initial layers, where token-to-token relationships are comparatively simpler, and learnable MHSA could be an *overkill*.

Following our previous nomenclature, $[G_nB]_{l_1,...,l_j}$ and $[G_nU]_{l_1,...,l_j}$ denote the blockwise and universal Fourier KArAt variants, respectively, where $l_1, \ldots, l_j$ layers use learnable attention and the rest $(L - j)$ layers use traditional `softmax`. Using Fourier KArAt only in the final layer of ViT-Base, -Small, -Tiny, each with 12 encoder layers, not only decreases the overall parameters and enables more tractable training, but it also improves the performance; see Tables 5 and 6.

(*ii*) **Attention Sparsification** explores effective sparsification strategies for calculating a sparse attention matrix for efficient implementation of attention mechanisms in transformers used for visual understanding Rahimian et al. (2024); Zhang & Gong (2023) or language modelling Zaheer et al. (2020); Shi et al. (2021); Guo et al. (2019) tasks. Sparse attentions Yun et al. (2020); Kovaleva et al. (2019); Zhang et al. (2021) conventionally use a support mask to determine which values in an attention matrix to consider. Motivated by this, we use an attention masking strategy to sparsify the KArAt activated matrices.

Since Fibottention uses a structured sparsity that varies across attention heads, it reduces redundant pairwise interactions in image tokens for visual understanding tasks and obtains comparable performances Rahimian et al. (2024). We use it for sparsification. In particular, we use element-wise multiplication of the support mask on every attention matrix after it is activated by KArAt. For any ViT Fourier KArAt, blockwise $G_nB$ or universal $G_nU$, with grid size $n$, we denote their sparsified versions as $G_nB^\dagger$ and $G_nU^\dagger$, respectively. We observe that sparsification helps the models achieve better performance; see Table 5.

**Performance Improvement.** From Table 5, we observe ViT-Base, -Small, and -Tiny $+[G_3B]_{12}$ (Fourier KArAt in the 12th layer, others are `softmax`) outperform the traditional MHSA on CIFAR-10 by $\sim 4 - 7\%$, and on CIFAR-100 by approximately $\sim 10 - 16\%$, respectively. On ImageNet-1K, the accuracy drop remains less than a modest $0.78\%$ for all three models, which is a vast improvement from the results in Table 1. From Table 6, for transfer learning, the $+[G_3B]_{12}$ models outperform vanilla ViT-Base on SVHN, ViT-Small on SVHN and Flowers 102, and ViT-Tiny on Flowers 102 and STL-10, compared to the standard Fourier KArAt that could not outperform any. Attention sparsification Rahimian et al. (2024) improves the performance of the Fourier KArAt variants, and outperforms the ViT-Base, -Small, and -Tiny with traditional attention by $\sim 2 - 9\%$ on CIFAR-10, and $\sim 9 - 13\%$ on CIFAR-100. These results show that Fourier KArAt can be made scalable for larger models by using attention sparsification or parameter reduction. We note that these solutions towards scalability are not unique or the most optimal. We encourage the researchers to investigate KArAt with better strategies for improved parameter reduction and enhanced training stability.

## 8 CONCLUSION

Across our benchmarking, KArAt mostly outperforms traditional MHSA in ~~smaller~~ ViTs; ~~results on larger models are mixed.~~ Additionally, we note that the computing requirement impedes KArAt's performance—training and fine-tuning these models are expensive. However, learnable activations can make a substantial difference in a model's interpretability, and these limitations are not KArAt's weaknesses. At its early stage, KArAt's decent performance on diverse tasks shows its potential. In the future, one can extend this study beyond ViTs and check the resilience of KArAt in language processing and on large multimodal models AI; Yang et al. (2024); Thawakar et al. (2025); Campos et al. (2026); Brown et al. (2020); AI (2024); Team et al. (2023); Alayrac et al. (2022).

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

CONTENTS

APPENDIX

**Organization.** We organized the Appendix as follows: §B contains the solution to problem equation 3. We also provide the pseudocode to implement Fourier-KArAt in ViT's encoder block; see Algorithm 2. §C discusses additional numerical results, including implementation details, hyperparameter tuning, computation time analysis, diverse ablation studies, and different variants of Fourier KArAt. Finally, we concluded §C with the spectral analysis of the attention matrices for traditional and learnable attention.

## A    MULTIHEAD SELF ATTENTION—CONTINUED

Let $X \in \mathbb{R}^{H \times W \times C}$ be an input image of resolution $H \times W$. ViT uses a $p \times p$ patch and generates $N = \frac{HW}{p^2}$ input tokens. Each token is vectorized along $C$ channels to produce $X_T \in \mathbb{R}^{N \times p^2 C}$, where each row, $X_{T i,:}$, represents a flattened patch across channels. A learnable embedding matrix, $E \in \mathbb{R}^{p^2 C \times d}$ projects $X_{T i,:}$ to an embedding vector in $\mathbb{R}^d$ such that after appending the learnable class token, $x_{\mathcal{G}}$ and adding the positional encoding matrix, $P_E \in \mathbb{R}^{(N+1) \times d}$ of compatible size, the final input, $X_I = [x_{\mathcal{G}}; X_E] + P_E$. The matrix $X_I$ is layer normalized and further projected on the row spaces of three learnable weight matrices, $W_Q, W_K, W_V \in \mathbb{R}^{d \times d}$ to generate the query, key, and value matrices, $Q, K$, and $V$, respectively, via $Q = X_I W_Q, K = X_I W_K$, and $V = X_I W_V$. These matrices are further divided into $h$ partitions (also, called *heads*), $Q^i, K^i, V^i \in \mathbb{R}^{N \times \frac{d}{h}}$, such that each partition generates a self-attention matrix for the $i^{\text{th}}$ head, $\mathcal{A}^i = \frac{Q^i K^{i\top}}{\sqrt{d/h}}$. Finally, we project $V^i$ onto the column space of $\sigma(\mathcal{A}^i)$ as $\sigma(\mathcal{A}^i) V^i$, where $\sigma$ is the `softmax` operator applied row-wise. The outputs of different heads are further concatenated into a large matrix $\left[\sigma(\mathcal{A}^1) V^1 \quad \sigma(\mathcal{A}^2) V^2 \quad \cdots \quad \sigma(\mathcal{A}^h) V^h\right] W^O$, post-multiplied by the learnable weight, $W^O \in \mathbb{R}^{d \times d}$. This process is sequentially performed in $L$ encoder blocks; see Figure 1b(i).

## B    A SPARSE APPROXIMATION PROBLEM AND ITS SOLUTION

Let $y \in \mathbb{R}^m$ be a given vector. If we want to approximate $y$ with a vector, $x \in \mathbb{R}^m$, with positive components and $\|x\|_1 = 1$, we can write the *constrained optimization* problem as:

$$\begin{cases} x^\star = \arg\min_{x \in \mathbb{R}^m} \frac{1}{2}\|x - y\|^2 \\ \text{subject to } x_i \geq 0 \text{ for } i \in [m], \text{ and } \|x\|_1 = 1. \end{cases} \tag{3}$$

First, we rewrite the constrained problem equation 3 as an unconstrained problem using the Lagrange multipliers as:

$$\mathcal{L}(x, \lambda, \mu) = \frac{1}{2}\|x - y\|^2 + \lambda \left( \sum_{i=1}^{m} x_i - 1 \right) - \mu^T x, \tag{4}$$

where $\lambda, \mu$ are Lagrange multipliers. Using the Karush-Kuhn-Tucker (KKT) stationarity condition on equation 4, we find $0 \in \partial \mathcal{L}(x, \lambda, \mu)$, which implies, $x_i - y_i + \lambda = \mu_i$. The complementary slackness gives, $\mu_i x_i = 0$ for $i \in [m]$. Further, for $i \in [m]$, the primal and dual feasibility conditions are $x_i \geq 0$, $\sum_{i=1}^{m} x_i = 1$, and $\mu_i \geq 0$, respectively.

The stationarity and complementary slackness conditions give, $x_i - y_i + \lambda \geq 0$, $(x_i - y_i + \lambda) x_i = 0$. If $\lambda \geq y_i$, then $x_i = 0$. Otherwise, we have that $\sum_{i=1}^{m} x_i = \sum_{i=1}^{m} (y_i - \lambda) = 1$ implying $\lambda = \frac{1}{m}(\sum_{i=1}^{m} y_i - 1)$.

---

**Algorithm 1** Projection on $\ell_1$ ball Condat (2016)

---

**Input:** $y \in \mathbb{R}^m$

**Sort:** $y$ as $y_{(1)} \geq y_{(2)} \geq \ldots \geq y_{(m)}$

**Calculate:** $\rho \triangleq \max \left\{ i \in [m] \mid y_{(i)} - \frac{1}{i} \left( \sum_{j=1}^{i} y_{(j)} - 1 \right) \right\}$

**Set:** $\lambda = \frac{1}{\rho} \left( \sum_{i=1}^{\rho} y_{(i)} - 1 \right)$

**Output:** $x^\star = (y - \lambda)_+ \triangleq \max(y - \lambda, 0)$

---

---

**Algorithm 2** Fourier-KArAt in $j^{\text{th}}$ Vision Transformer block

---

**Input:** $X \in \mathbb{R}^{N \times d}$, Fourier basis $\{\sin(\cdot), \cos(\cdot)\}$

**Output:** $O \in \mathbb{R}^{N \times d}$

**Parameters:** $W_Q^i, W_K^i, W_V^i \in \mathbb{R}^{d \times d_h}, d_h = \frac{d}{h}, G, [\{[a_{pqm}], [b_{pqm}]\}_{m=1}^{G}, W^{i,j}], W^{O,j}$

**Hyperparameters:** Blockwise, universal, **Algorithm** 1 ($\ell_1$ projection), $L$ encoder blocks

**for each** head $i \in \{1, \ldots, h\}$ **do**:

    $Q^{i,j} \leftarrow X W_Q^{i,j}$

    $K^{i,j} \leftarrow X W_K^{i,j}$

    $V^{i,j} \leftarrow X W_V^{i,j}$

    $\mathcal{A}^{i,j} = \frac{Q^{i,j}(K^{i,j})^\top}{\sqrt{d_h}}$

**end for**

**for each** head $i \in \{1, \ldots, h\}$ **do**:

    **for each** row $k \in \{1, \ldots, N\}$ **do**:

        $\widehat{\Phi}^{i,j}[(\mathcal{A}^{i,j})_{k,:}^\top] \leftarrow \sum_{q=1}^{N} \widehat{\phi}_{pq}^{i,j}(\mathcal{A}_{k,q}^{i,j}) = \sum_{m=1}^{G} a_{pqm} \cos(m\mathcal{A}_{k,q}^{i,j}) + b_{pqm} \sin(m\mathcal{A}_{k,q}^{i,j})$

        **if** $\ell_1$ projection **then**

            Execute **Algorithm** 1

        **else**

            **pass**

        $\widehat{\sigma}(\mathcal{A}_{k,:}^{i,j}) \leftarrow \left[ W^{i,j} \widehat{\Phi}^{i,j}[(\mathcal{A}_{k,:}^{i,j})^\top] \right]^\top$

    **end for**

**end for**

$O \leftarrow \begin{bmatrix} \widehat{\sigma}(\mathcal{A}^{1,j})V^{1,j} & \widehat{\sigma}(\mathcal{A}^{2,j})V^{2,j} & \cdots & \widehat{\sigma}(\mathcal{A}^{N,j})V^{N,j} \end{bmatrix} W^{O,j}$

**if** Blockwise Mode **then**

    Return $O$

**else if** universal Mode **then**

    pass learnable units and parameters to $(j+1)^{\text{th}}$ encoder block

    Return $O$

---

Note that, in Algorithm 1, one can replace $y$ with $\widehat{\Phi}^{i,j}[(\mathcal{A}^{i,j})_{k,:}^\top]$ so that the constraint $\|\widehat{\sigma}\left[(\mathcal{A}_{k,:}^{i,j})\right]\|_1 = 1$ and $\widehat{\sigma}\left[(\mathcal{A}_{k,l}^{i,j})\right] \geq 0$ is satisfied for the low-rank attention matrices in Fourier-KArAt. This algorithm is also kept as an optional sub-process for Algorithm 2.

## C  ADDENDUM TO EMPIRICAL STUDY

This section complements our empirical results in §5.

### C.1  IMPLEMENTATION DETAILS

We implement the framework in Python using PyTorch Paszke et al. (2019) and use the training strategy of DEiT Touvron et al. (2021). We train all models for 100 epochs with the ADAM optimizer Kingma & Ba (2015) with a base learning rate of $3.125 \times 10^{-5}$ and batch-size 32, except for the experiments on ImageNet-1K that use a learning rate of $1.25 \times 10^{-4}$ with batch-size 128. All experiments use a warm-up of 5 epochs and a cosine scheduler with a weight decay of $5 \times 10^{-2}$. The experiments were performed on two 80 GB NVIDIA H100 GPUs. The hyperparameter settings for experiments with ViTs are given in Table 7.

Table 7: **Hyper-parameter settings** for image classification and recognition experiments conducted in this work for vanilla ViTs and ViT+Fourier KArAts.

| | |
|---|---|
| Input Size | $224 \times 224$ |
| Crop Ratio | 0.9 |
| Batch Size | 128 for ImageNet-1K and 32 for CIFAR-10 & CIFAR-100 |
| Optimizer | AdamW |
| Optimizer Epsilon | $1 \times 10^{-6}$ |
| Momentum | 0.9 |
| Weight Decay | 0.05 |
| Gradient Clip | 1.0 |
| Learning Rate Schedule | Cosine |
| Learning Rate | $5 \times 10^{-4} \times \frac{\text{Batch Size}}{512}$ |
| Warmup LR | $1 \times 10^{-6}$ |
| Min LR | $1 \times 10^{-5}$ |
| Epochs | 100 |
| Decay Epochs | 1 |
| Warmup Epochs | 5 |
| Decay Rate | 0.988 |
| Exponential Moving Average (EMA) | True |
| EMA Decay | 0.99992 |
| Random Resize & Crop Scale and Ratio | $(0.08, 1.0), (0.67, 1.5)$ |
| Random Flip | Horizontal 0.5; Vertical 0.0 |
| Color Jittering | 0.4 |
| Auto-augmentation | rand-m15-n2-mstd1.0-inc1 |
| Mixup | True |
| Cutmix | True |
| Mixup, Cutmix Probability | 0.5, 0.5 |
| Mixup Mode | Batch |
| Label Smoothing | 0.1 |
| Patch Size | 16 |

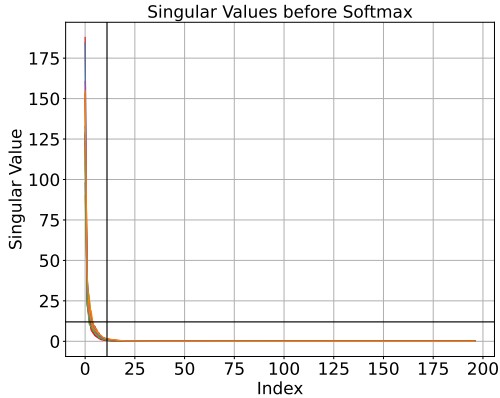 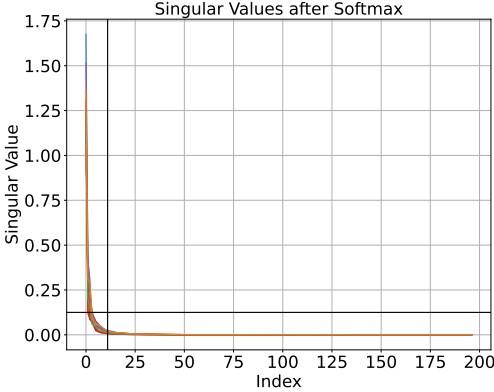

(a) Attention matrix $\mathcal{A}^{i,j}$ before `softmax` activation.

(b) Attention matrix $\sigma(\mathcal{A}^{i,j})$ after `softmax` activation.

Figure 5: **Spectral analysis** of the attention matrices before and after `softmax` shows that they are low-rank. For this experiment, we use all 3 heads in the last encoder block of ViT-Tiny on 5 randomly sampled images from the CIFAR-10 validation set. We plot all 15 singular vectors (each of 197 dimensions) where the singular values are arranged in non-increasing order.

### C.1.1 BASELINES AND DATASETS

For our image classification benchmarking, we chose 5 popular vision Transformers—ViT-Tiny, ViT-Small Wu et al. (2022); Steiner et al. (2022), ViT-Base Dosovitskiy et al. (2020), ConViT-Tiny D'Ascoli et al. (2021), and Swin-Transformer-Tiny Liu et al. (2021). We incorporate Fourier-KArAt in them by replacing their `softmax` function with a learnable activation. We perform our benchmarking on CIFAR-10, CIFAR-100 Krizhevsky et al. (2009) and ImageNet-1K Deng et al. (2009) datasets. We use small-scale datasets, SVHN Netzer et al. (2011), Oxford Flowers-102 Nilsback & Zisserman (2008), and STL-10 Coates et al. (2011) for understanding the transfer learning capability of the models. Additionally, we use ViT-Det Li et al. (2022) on the MS COCO Lin et al. (2014) dataset for the object detection and segmentation task.

### C.1.2 SMALL-SCALE DATASETS USED

We use 3 small-scale datasets, SVHN Netzer et al. (2011), Oxford Flowers 102 Nilsback & Zisserman (2008), and STL-10 Coates et al. (2011) for transfer learning task. SVHN is similar to MNIST LeCun et al. (2010), a 10-class digit classification dataset with small images like CIFAR-10 Krizhevsky et al. (2009), divided into 73,257 training and 26,032 test images. The challenge in this dataset comes from the distracting elements surrounding the concerned class. The OxfordFlowers-102 Nilsback & Zisserman (2008) is a small yet challenging dataset with unbalanced training images (40-254 images) for the 102 classes. The STL-10 Coates et al. (2011) is a CIFAR-like 10-class dataset with a larger image size, but only 500 training and 800 testing images per class.

### C.1.3 IMPLEMENTING DIFFERENT BASIS AND BASE ACTIVATION FUNCTIONS IN KARAT

Theorem 2 uses the Fourier basis to span any $L^p$ periodic function, where $p \in (1, \infty]$. With this view, we aim to represent the `softmax` function using any chosen set of basis functions. Although our KArAt design spans each attention unit as a linear combination of Fourier basis, the originally designed KAN layer Liu et al. (2025) includes a *residual activation base* function, $b(x)$, which was set to `SiLU`; any other activation function can be used. We refer to $b(x)$ as the *base activation function*. To see how these base functions influence the performance of Fourier KArAt, we run both blockwise and universal modes on ViT-Tiny+KArAt with `Identity`[1], `SiLU`, and `GELU`[2] base functions. Similar to equation 2, the attention unit then has the following form:

$$\widehat{\phi}_{pq}^{i,j}(\mathcal{A}_{k,q}^{i,j}) = b(\mathcal{A}_{k,q}^{i,j}) + \sum_{m=1}^{G} a_{pqm} \cos\left(m\mathcal{A}_{k,q}^{i,j}\right) + b_{pqm} \sin\left(m\mathcal{A}_{k,q}^{i,j}\right). \tag{5}$$

---

[1] `Identity`$(t) = t$.

[2] `GELU`$(t) = t\psi(t)$, where $\psi(\cdot)$ denotes the cumulative distribution function for the Gaussian Distribution.

For the experiments in the main paper, in the general form, equation 5, we use $b(x) = 0$ for all $x \in \mathbb{R}$, grid size, $G = 3$, and the row dimension of the $\widehat{\Phi}^{i,j}$ is set to $r = 12$, or otherwise noted. We encourage the reader to explore other configurations that differ from those previously outlined.

(*i*) **Different Basis Functions Used and Their Results.** We incorporate a variety of widely used KAN activation functions into KArAt. By design, KArAt can utilize *any basis function* for activating the attention units. Therefore, in addition to the Fourier basis, we embed 5 different wavelet bases, and the Rational Function basis into KArAt. The function representations for these choices of bases are outlined in Table 8.

We use the safe Padé approximation unit (PAU) of order $(m, n)$ for rational function basis; KAT also uses the same basis function Yang & Wang (2025). In contrast to PAUs, safe PAUs ensure stability and the prevention of poles, making it a clear choice for rational functions. We only consider safe PAUs of order $(5, 4)$ in this paper; other order combinations can be used.

Being an integral part in signal analyses and image reconstruction, wavelet functions serve as a candid choice for approximating signals and feature extraction Bozorgasl & Chen (2024). Having zero mean and finite energy drives wavelets to make fair and meaningful representations of signals in local regimes. Similar to the Fourier transform, the inverse continuous wavelet transform (CWT) resolves the original signal. We consider mother wavelets, mentioned in Wav-KAN Bozorgasl & Chen (2024), as basis functions. Unique to this choice, KArAt learns scale, $s$, translation, $\tau$, as measured in the CWT, and a learnable weight coefficient $w$. For any choice of wavelet base function, $\phi$, the input is first shifted by the scale and translation parameters and then activated by the mother wavelet.

Embedding different bases in other ViT variants with KArAt is straightforward. Although our original design in Algorithm 2 represents each matrix entry $\widehat{\phi}_{pq}$ using the Fourier basis, any functions outlined in Table 8 can be used. All other components in Algorithm 2 stay the same.

We implement rational and wavelet basis activation functions from Table 8 in ViT-Tiny+KArAt and ViT-Base+KARAT using the *blockwise* configuration and show their performance on the image classification task on CIFAR-10 and CIFAR-100; see the results in Table 10. Performance across model variants is not consistent with the choice of wavelet basis function used. ViT-Tiny+KArAt achieves its highest Top-1 accuracy with DOG (derivative of Gaussian) on CIFAR-10 and -100, whereas Rational (5,4) breaks this trend with ViT-Base+KArAt. Another observation shows that ViT-Base+KArAt performs poorly with both Meyer and Morlet functions, contrary to ViT-Tiny+KArAt. Hence, we cannot make a proper remark regarding the choice of basis.

(*ii*) **Base Activation Functions Used and Their Results.** We implement different base functions in ViT-Tiny+KArAt using the *blockwise* and *universal* configurations and show their performance on the image classification task on CIFAR-10; see the results in Table 9. For the majority of our experiments, including those in the main paper, we define and use `ZeroModule` and set the base activation function to be identically zero, or $b(x) = 0$. Slightly lower computations within a single KAN layer help motivate us to use this base activation function. Overall, choosing between the activations `SiLU` and `GELU` for ViT-Tiny+KArAt with $G_3B, G_3U$ modes results in a slightly lower Top-1 accuracy compared to the results using the `ZeroModule` in the main paper (see Table 1).

### C.1.4 DEPLOYING FOURIER KARAT TO OTHER ViTs

We consider three modern ViT-based architectures, including ConViT D'Ascoli et al. (2021) and Swin Transformer Liu et al. (2021). Incorporating the Fourier KArAt variants into these models is not straightforward. Below, we outline their implementation details.

(*i*) **ConViT D'Ascoli et al. (2021)** uses two different attention mechanisms, the gated positional self-attention (GPSA) and MHSA. All of the variants of ConViT vary the number of attention heads per encoder while keeping the number of encoders the same. The purpose of the ConViT is to combine a CNN and a ViT in one encoder block. With the first ten layers using GPSA and the last two layers using MHSA, the model learns a balanced structure between CNNs and ViTs. Our variant of ConViT replaces the MHSAs with Fourier KArAts. We implement the universal and blockwise attention modes on this model. We give the ConViT's hyperparameter configuration details in Table 11.

(*ii*) **Swin Transformers Liu et al. (2021)** has a variable number of tokens. The number of tokens considered in the attention operation varies based on the window size, and thus, we do not implement a universal attention mode due to incompatibility. We replace regular and shifted window MHSAs, W-MHSA and SW-MHSA, with regular and shifted Fourier KArAts. We give the Swin-Transformer's hyperparameter configuration details in Table 12.

Table 8: **Different basis functions and their representations for** $b(x) = 0$. * For our experiments, the morlet central frequency hyperparameter is $\omega_0 = 5$, but other nonnegative values can be used. ** Meyer is defined to be $(m \circ \nu)(\tilde{x}) = m(\nu(\tilde{x}))$, where $m(t) = \mathbb{I}\left(-\infty, \frac{1}{2}\right](t) + \mathbb{I}\left(\frac{1}{2}, 1\right)(t)\cos\left(\frac{\pi}{2}\nu(2t-1)\right)$ and $\nu(x) = x^4(35 - 84x + 70x^2 - 20x^3)\mathbb{I}[0,1](x)$.

| Basis | Function Representation, $\phi(x)$ | Initialized Specifications |
|---|---|---|
| Fourier | $\sum\limits_{k=1}^{G}\left(a_k\cos\left(kx\right) + b_k\sin\left(kx\right)\right)$ | $a_k,\ b_k \sim \mathcal{N}(0,1)$, $G$ denotes the grid size |
| Rational $(m, n)$ | $\frac{a_0 + a_1 x + \cdots + a_m x^m}{1 + \lvert b_1 x + \cdots + b_n x^n \rvert}$ | $a_i,\ b_j \sim \mathcal{N}(0,1),\ i = 0,\ldots,m,\ j = 1,\ldots,n$ |
| Mexican Hat | $\frac{2w}{\pi^{1/4}\sqrt{3}}\left(\tilde{x}^2 - 1\right)e^{-\frac{\tilde{x}^2}{2}},\ \tilde{x} = \frac{x-\tau}{s}$ | |
| Morlet* | $w\cos\left(\omega_0\tilde{x}\right)e^{-\frac{\tilde{x}^2}{2}},\ \tilde{x} = \frac{x-\tau}{s}$ | |
| DOG | $-w\frac{d}{dx}\left(e^{-\frac{\tilde{x}^2}{2}}\right),\ \tilde{x} = \frac{x-\tau}{s}$ | $w \sim \mathcal{N}(0,1)$, (Translation) $\tau = 0$, (Scale) $s = 1$ |
| Meyer** | $w\sin\left(\pi\lvert\tilde{x}\rvert\right)(m \circ \nu)(\tilde{x}),\ \tilde{x} = \frac{x-\tau}{s}$ | |
| Shannon | $w\,\mathrm{sinc}\left(\frac{\tilde{x}}{\pi}\right)\omega(\tilde{x}),\ \tilde{x} = \frac{x-\tau}{s}$ | $\omega(\tilde{x})$ is the symmetric hamming window |

Table 9: **Blockwise** and **universal** ViT-Tiny+KArAt and their Top-1 accuracies on CIFAR-10 using different base activation functions (`SiLU`, `Identity`, `GELU`).

| Model | Base Activation | Acc.@1 on CIFAR-10 |
|---|---|---|
| ViT-Tiny+$G_3B$ | Identity | 75.90 |
| | SiLU | 76.25 |
| | GELU | 76.48 |
| ViT-Tiny+$G_3U$ | Identity | 75.21 |
| | SiLU | 75.25 |
| | GELU | 75.28 |
| ViT-Tiny+$G_1B$ | Identity | 74.87 |
| | SiLU | 75.53 |
| | GELU | 75.18 |
| ViT-Tiny+$G_1U$ | Identity | 74.44 |
| | SiLU | 75.12 |
| | GELU | 74.82 |

Table 10: **Blockwise configuration** ViT+KArAt and their Top-1 accuracies using different base functions on CIFAR-10 and CIFAR-100.

| Model | Basis Function | Acc.@1 | |
|---|---|---|---|
| | | CIFAR-10 | CIFAR-100 |
| ViT-Tiny+KArAt | Rational (5,4) | 69.22 | 37.93 |
| | Mexican Hat | 69.62 | 38.92 |
| | Morlet | 70.84 | 37.88 |
| | DOG | 73.89 | 44.23 |
| | Meyer | 71.13 | 41.65 |
| | Shannon | 71.41 | 39.40 |
| ViT-Base+KArAt | Rational (5,4) | 72.59 | 41.52 |
| | Mexican Hat | 72.58 | 44.44 |
| | Morlet | 24.19 | 6.22 |
| | DOG | 72.28 | 48.69 |
| | Meyer | 28.83 | 6.67 |
| | Shannon | 69.84 | 39.98 |

### C.1.5 IMPLEMENTING FOURIER KARAT FOR OBJECT DETECTION AND SEMANTIC SEGMENTATION

Implementing Fourier KArAt for detection and segmentation is an involved task. Firstly, RCNN He et al. (2017); Girshick (2015); Ren et al. (2016) frameworks employ an augmentation strategy to train on dynamic image sizes, which also dynamically change the number of tokens and thus, token interactions in the attention. However, KArAt needs to ensure fixed-length input and output; hence, the dynamic resize becomes incompatible. Secondly, vanilla ViT-Det Li et al. (2022) involves window partitioning in the self-attention, making the number of tokens variable in the attention matrices throughout the blocks of encoders. Thirdly, as this window mechanism changes the number of token interactions in the attention matrices, the universal mode of KArAt also becomes incompatible. To solve all these, we restrict the input image size to $224 \times 224$, and use a window size of $14$ and a patch size of $16$. This inherently fixes the number of tokens to $196$, and the window size of $14$ ensures $196$ tokens in the windowed attention blocks. Thus, the number of tokens throughout the model remains fixed. We note that the restriction on augmentation and limiting the input size impacts the baseline performance of ViT-Det. The performance analysis is detailed in §5.1 and inference visualization is provided in Figure 8.

### C.1.6 PLOTTING LOSS LANDSCAPES

Following Li et al. (2018), we perform principal component analysis (PCA) of the parameter change over training progression to understand the major directions of parameter convergence. Considering

Table 11: **Hyper-parameter settings** for image classification conducted in this work for vanilla ConViT-Tiny and ConViT-Tiny+Fourier KArAt.

| | |
|---|---|
| Input Size | $224 \times 224$ |
| Batch Size | 128 for Imagenet-1K & 32 for CIFAR-10 |
| Optimizer | AdamW |
| Optimizer Epsilon | $1 \times 10^{-8}$ |
| Momentum | 0.9 |
| Weight Decay | 0.05 |
| Gradient Clip | 1.0 |
| Learning Rate Schedule | Cosine |
| Learning Rate | $5 \times 10^{-4} \times \frac{\text{Batch Size}}{512}$ |
| Warmup LR | $1 \times 10^{-6}$ |
| Min LR | $1 \times 10^{-5}$ |
| Epochs | 100 |
| Decay Epochs | 30 |
| Warmup Epochs | 5 |
| Decay Rate | 0.1 |
| Exponential Moving Average (EMA) | False |
| EMA Decay | 0.99996 |
| Locality up to Layer | 10 |
| Locality Strength | 1 |
| Random Resize & Crop Scale and Ratio | $(0.08, 1.0), (0.67, 1.5)$ |
| Random Flip | Horizontal 0.5; Vertical 0.0 |
| Color Jittering | 0.4 |
| Auto-augmentation | rand-m9-mstd0.5-inc1 |
| Mixup | True |
| Cutmix | True |
| Mixup, Cutmix Probability | 0.5, 0.5 |
| Mixup Mode | Batch |
| Label Smoothing | 0.1 |

Table 12: **Hyper-parameter settings** for image classification conducted in this work for vanilla Swin-Transformer-Tiny and Swin-Transformer-Tiny+Fourier KArAt.

| | |
|---|---|
| Input Size | $224 \times 224$ |
| Crop Ratio | 0.9 |
| Batch Size | 128 for Imagenet-1K & 32 for CIFAR-10 |
| Optimizer | AdamW |
| Optimizer Epsilon | $1 \times 10^{-8}$ |
| Momentum | 0.9 |
| Weight Decay | 0.05 |
| Gradient Clip | 1.0 |
| Learning Rate Schedule | Cosine |
| Learning Rate | $5 \times 10^{-4} \times \frac{\text{Batch Size}}{512}$ |
| Warmup LR | $1 \times 10^{-6}$ |
| Min LR | $1 \times 10^{-5}$ |
| Epochs | 100 |
| Decay Epochs | 30 |
| Patience Epochs | 10 |
| Warmup Epochs | 5 |
| Decay Rate | 0.1 |
| Exponential Moving Average (EMA) | False |
| EMA Decay | 0.99996 |
| Fused Layer Norm | False |
| Fused Window Process | False |
| Window Size | 7 |
| Random Resize & Crop Scale and Ratio | $(0.08, 1.0), (0.67, 1.5)$ |
| Random Flip | Horizontal 0.5; Vertical 0.0 |
| Color Jittering | 0.3 |
| Auto-augmentation | rand-m9-mstd0.5-inc1 |
| Mixup | True |
| Cutmix | True |
| Mixup, Cutmix Probability | 0.5, 0.5 |
| Mixup Mode | Batch |
| Label Smoothing | 0.1 |

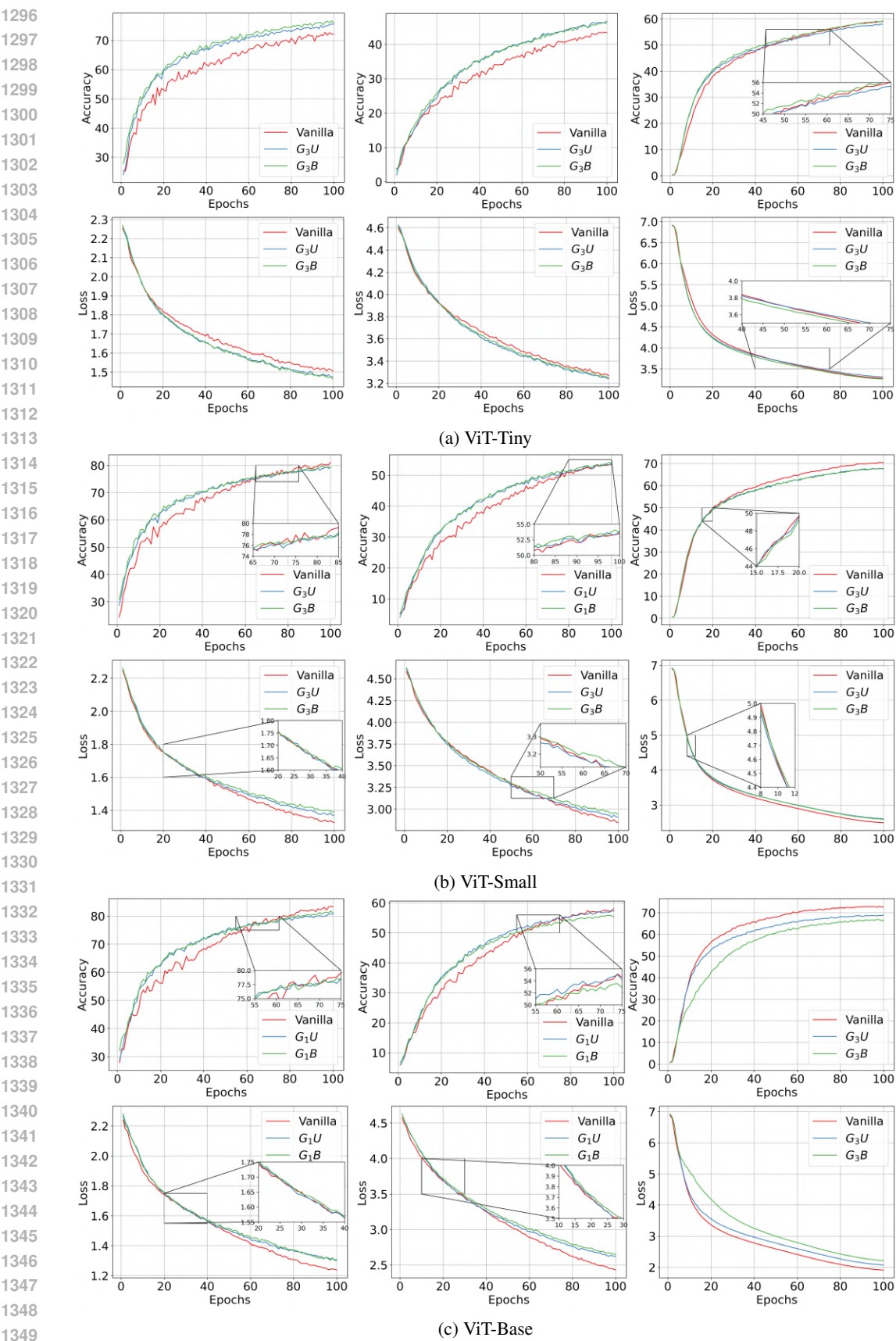

Figure 6: **Training loss and test accuracy** of vanilla ViTs and their Fourier KArAt versions on CIFAR-10, CIFAR-100, and ImageNet-1K datasets (left to right).

Table 13: **Hyper-parameter settings** for object detection and instance segmentation experiments.

| | |
|---|---:|
| Input Size | $224 \times 224$ |
| Batch Size | 32 |
| Optimizer | AdamW |
| Optimizer Epsilon | $1 \times 10^{-6}$ |
| Momentum | $0.9, 0.999$ |
| Weight Decay | 0.1 |
| Learning Rate Schedule | Warmup Scheduler |
| Learning Rate | $1 \times 10^{-4}$ |
| Iterations | 184375 |
| Decay Rate | 0.988 |
| Random Flip | Horizontal 0.5; Vertical 0.0 |
| Patch Size | 16 |
| Attention Window Size | 14 |
| Window Attention Block Indices | $0, 1, 3, 4, 6, 7, 9, 10$ |
| Encoder Output Layer Index | 11 |
| Pyramid Scale Factors | $4.0, 2.0, 1.0, 0.5$ |
| Output Channels | 256 |
| Proposal Generator Input Layers (corresponding to feature pyramid) | $p2, p3, p4, p5, p6$ |
| Proposal Generator Input Sizes | $32, 64, 128, 256, 512$ |
| Proposal Generator Training Pre-NMS Top-K | 12000 |
| Proposal Generator Evaluation Pre-NMS Top-K | 6000 |
| Proposal Generator Training Post-NMS Top-K | 2000 |
| Proposal Generator Evaluation Post-NMS Top-K | 1000 |
| Proposal Generator NMS Threshold | 0.7 |
| Region-of-Interest Heads IOU Threshold | 0.5 |
| Region-of-Interest Score Threshold | 0.05 |
| Region-of-Interest NMS Threshold | 0.5 |

Table 14: Fourier KArAt on object detection and instance segmentation tasks on the MS COCO Lin et al. (2014) dataset. The header *Box* and *Mask* refer to detection and segmentation tasks, respectively.

| Model | Initialization | Box | | | Mask | | |
|---|---|---|---|---|---|---|---|
| | | AP | $AP_{50}$ | $AP_{75}$ | AP | $AP_{50}$ | $AP_{75}$ |
| ViT-Det-Base | | 32.34 | 49.16 | 34.43 | 28.35 | 46.04 | 29.48 |
| + $G_1 B$ | ViT-Base on ImageNet-1K | 22.48 | 36.82 | 23.13 | 20.11 | 34.09 | 20.46 |
| + $G_1 U$ | | 26.68 | 43.25 | 27.88 | 24.01 | 40.21 | 24.66 |
| ViT-Det-Base | | 15.28 | 26.88 | 15.22 | 13.39 | 24.34 | 12.86 |
| + $G_1 B$ | Random Initialization | 10.32 | 18.77 | 10.05 | 8.94 | 16.51 | 8.50 |
| + $G_1 U$ | | 10.99 | 20.04 | 10.82 | 9.69 | 17.99 | 9.25 |

the fully trained model as the minimum in the loss hyperplane and the two principal component directions as $X$ and $Y$ axes, we plot the loss values over the validation set of CIFAR-10 along the $Z$ axis for ViT-Tiny and -Base for the traditional MHSA and KArAt ($G_3 B$ and $G_1 B$, respectively).

## C.2 *Generalizability*: DETECTION & SEGMENTATION—CONTINUED

We choose Mask RCNN He et al. (2017)-based framework for this purpose and employ a feature pyramid network Lin et al. (2017) based on ViT-Det Li et al. (2022); it is non-trivial to implement Fourier KArAt, see §C.1.5. We trained ViT-Det Li et al. (2022) using ViT-Base backbone on the MS COCO Lin et al. (2014) dataset for 50 epochs with two settings: (*i*) fine-tuning on the ImageNet-1K pre-trained weights on traditional softmax attention, and (*ii*) random initialization. Table 14 shows that for the fine-tuning on ImageNet pre-trained weights, Fourier-KArAt shows $\sim 4 - 13\%$ gap in average precision (AP) from its vanilla variant; their qualitative performance in Figure 8 is similar. Overall, for ViT-Det, Fourier-KArAt performs inferiorly to its conventional counterpart in detection and segmentation. Interestingly, Figure 9 shows, vanilla ViT-Det using softmax activation reaches its peak performance quickly (within 12 epochs), Fourier-KArAt delays in achieving it (within 30 and 45 epochs for $G_1 B$ and $G_1 U$, respectively), proving the incompatibility of softmax-based weights initialization. It is, however, still a good initialization for KArAt as the performance is better than random initialization.

## C.3 COMPUTATION TIME, FLOPS, MEMORY REQUIREMENT, AND THROUGHPUT OF FOURIER KARAT

The overall computation for Fourier KArAt variants is higher than their conventional softmax MHSA, and we have delineated it in Figure 7. Primarily, the Fourier KArAt variants have a longer training time. We show the training time comparison between the traditional MHSA and its Fourier KArAt versions for 100 epochs on CIFAR-10, CIFAR-100, and ImageNet-1K datasets for all the models (ViT-Tiny, -Small, and -Base) in Figures 7a– 7c, respectively. We only compare the best-performing Fourier KArAt models ($G_1 B$ and $G_1 U$ for ViT-Base, and $G_3 B$ and $G_3 U$ for ViT-Tiny & ViT-Small) with their traditional softmax MHSA counterparts. We also observe that universal mode $G_n U$ training times are consistently slightly less than the blockwise modes $G_n B$.

During the training, we monitored the GPU memory requirements, and as expected, Fourier KArAt variants utilize significantly more memory than traditional MHSA. In particular, the GPU memory requirements scale by $2.5 - 3\times$, compared to the traditional softmax MHSA.

We also compare the throughput during inference in Figure 7d and see slightly faster inference in universal mode than blockwise, except for ViT-Base. While there is a massive training time discrepancy between vanilla ViTs and Fourier KArAt ViTs, the inference speeds for Fourier KArAt variants are comparable to their vanilla counterparts. Although there is a minor difference in throughput between universal and blockwise modes during inference, theoretically, both variants for any model with the same grid size should have the same number of FLOPs.

## C.4 ABLATION STUDY WITH KAN AND FOURIER KARAT

In this section, we perform a detailed ablation study with different hyperparameter settings. Our first set of experiments shows why B-Splines are not a good basis for KAN for image classification tasks. The experiments with B-Splines solidify our argument for why we primarily use the Fourier basis in

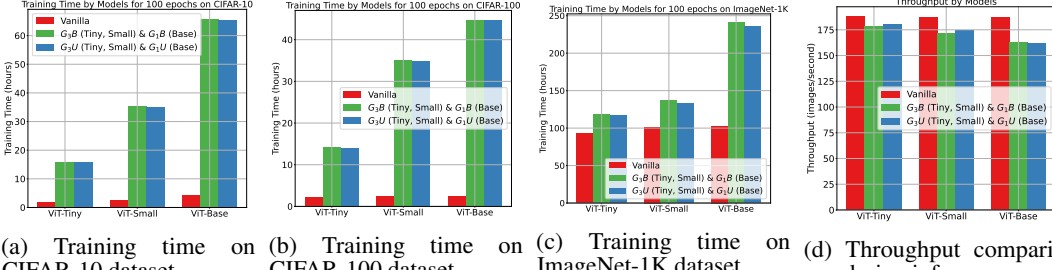

(a) Training time on CIFAR-10 dataset.
(b) Training time on CIFAR-100 dataset.
(c) Training time on ImageNet-1K dataset.
(d) Throughput comparison during inference.

Figure 7: **A detailed comparison of computing requirements.** We compare the training times for 100 epochs with the hyperparameter settings given in Table 7 for all the datasets CIFAR-10, CIFAR-100, and ImageNet-1K. We also compare the throughputs of different models on ImageNet-1K; the throughput results will be similar for other datasets, as the input size is $224 \times 224$.

Table 15: **Parameter, computation, and memory** requirement for Fourier-KArAt (with hidden dimension, $r = 12$) compared to the traditional `softmax` attention. This Table particularly shows the individual computation required for the attention activation. The memory requirement shown is approximate and is based on averages of batches of 32 images of resolution $224 \times 224$. We note that changing $r$ will affect the performance and memory requirements. In our main paper, all the experiments were performed with $r = 12$.

| Model | Parameters | | GFLOPs | | GPU Memory |
|---|---|---|---|---|---|
| | Attention Activation | Total | Attention Activation | Total | |
| ViT-Base | 0 | 85.81M | 0.016 | 17.595 | 7.44 GB |
| + $G_1B$ | 1.70M | 87.51M | 0.268 | 17.847 | 17.36 GB |
| + $G_1U$ | 0.14M | 85.95M | 0.268 | 17.847 | 16.97 GB |
| ViT-Small | 0 | 22.05M | 0.008 | 4.614 | 4.15 GB |
| + $G_3B$ | 1.53M | 23.58M | 0.335 | 4.941 | 11.73 GB |
| + $G_3U$ | 0.13M | 22.18M | 0.335 | 4.941 | 11.21 GB |
| ViT-Tiny | 0 | 5.53M | 0.005 | 1.262 | 2.94 GB |
| + $G_3B$ | 0.76M | 6.29M | 0.168 | 1.425 | 7.48 GB |
| + $G_3U$ | 0.06M | 5.59M | 0.168 | 1.425 | 7.29 GB |

KArAt. After that, we perform rigorous ablation studies of Fourier KArAt on the hidden dimension, grid size effects, and many others.

### C.4.1 EXPERIMENTS WITH KANS OPERATED ON B-SPLINE BASIS

We test the performance of the B-spline KANs in the classification task on the CIFAR-10 Krizhevsky et al. (2009), CIFAR-100 Krizhevsky et al. (2009), MNIST LeCun et al. (1998; 2010), and Fashion-MNIST Xiao et al. (2017) datasets. We performed extensive experiments to see if there are any benefits to choosing B-Splines for the basis functions in KAN layers. Table 16 shows the Top-1 accuracies from different variants of a Deep KAN. In this set of experiments, we closely followed earlier works Yu et al. (2024); Liu et al. (2025). Hyperparameters involved in these experiments include the order of the B-spline $k$, grid size $G$, grid range $[-I, I]$, layers, and width. Although the KANs with B-Spline basis yield high accuracies in the small-scale MNIST and Fashion-MNIST datasets, they fail to generalize over larger datasets (CIFAR-10 and -100).

### C.4.2 ABLATION WITH THE HIDDEN DIMENSION, $r$ IN FOURIER KARAT

While avoiding the computational overhead for computing $\Phi^{i,j} \in \mathbb{R}^{N \times N}$, we make use of the low-rank structure that the attention heads show (see Figure 5) by comparing different values of $r$. Particularly, we consider the values, $r = 8, 12, 24$, on the Fourier KArAt variant of ViT-Base model, $\widehat{\Phi}^{i,j} \in \mathbb{R}^{r \times N}$; see Table 18 for results on CIFAR-10. In this ablation, we observe that changing the hidden dimension has a negligible impact on the model's performance. This can be explained by the sudden drop in the singular values, as shown in Figure 13. As long as $r$ remains greater than the sudden drop index, the model should not be impacted by the changing of $r$ except for changes in

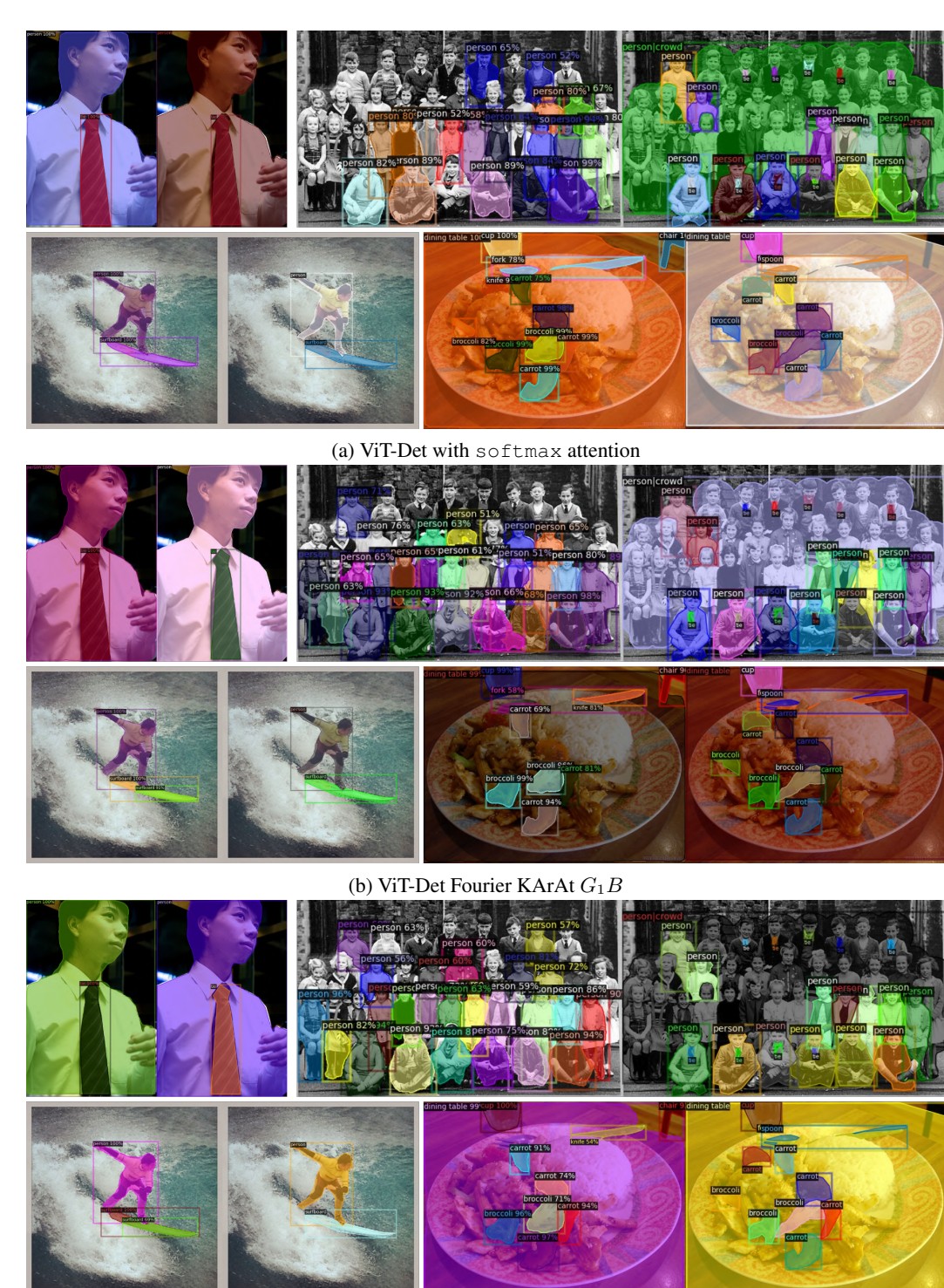

(a) ViT-Det with `softmax` attention

(b) ViT-Det Fourier KArAt $G_1 B$

(c) ViT-Det Fourier KArAt $G_1 U$

Figure 8: **Detection and segmentation tasks inference visualization** using ViT-Det with ViT-Base as backbone for traditional MHSA and Fourier KArAt. For each sample, the ground-truth is given on the right side and the inference is on the left.

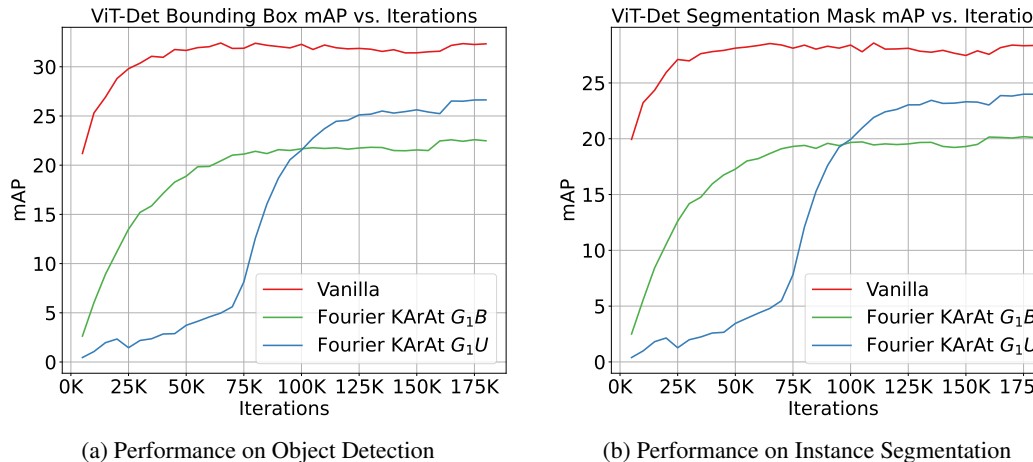

(a) Performance on Object Detection  (b) Performance on Instance Segmentation

Figure 9: **Training Curves of ViT-Det and Fourier KArAt** in object detection and instance segmentation tasks while training from ImageNet-1K pretrained ViT-Base with traditional MHSA weights initialization. In this particular task, the universal configuration ($G_1U$), performs strictly better than the blockwise configuration ($G_1B$).

Table 16: **Experimental results on multi-layer KANs organized similarly to an MLP.** These experiments involve B-Splines as the basis functions, as mentioned in Section 2.

(a) Experiments on CIFAR-10 and CIFAR-100.

| Spline Order | Grids | Grid Range | Layers | Acc.@1 CIFAR-10 | Acc.@1 CIFAR-100 |
|---|---|---|---|---|---|
| 1 | 10 | (-4, 4) | (10, 10) | 22 | 1 |
| | 20 | (-4, 4) | (10, 10) | 21 | 4 |
| | | (-6, 6) | (10, 10) | 22 | 4 |
| | | (-8, 8) | (10, 10) | 22 | 4 |
| | 40 | (-4, 4) | (10, 10) | 22 | 1 |
| | | (-6, 6) | (10, 10) | 22 | 1 |
| | | (-8, 8) | (10, 10) | 22 | 1 |
| 3 | 10 | (-4, 4) | (10, 10) | 22 | 1 |
| | 20 | (-4, 4) | (10, 10) | 22 | 4 |
| | | (-6, 6) | (10, 10) | 22 | 4 |
| | | (-8, 8) | (10, 10) | 39 | 4 |
| | 40 | (-4, 4) | (10, 10) | 22 | 1 |
| | | (-6, 6) | (10, 10) | 29 | 1 |
| | | (-8, 8) | (10, 10) | 31 | 1 |
| 4 | 10 | (-4, 4) | (10, 10) | 21 | 1 |
| 5 | 10 | (-1, 1) | (10, 10) | 10 | 4 |
| | | (-2, 2) | (10, 10) | 10 | 1 |
| | | (-4, 4) | (10, 10) | 22 | 4 |
| | | (-4, 4) | (20, 20) | 40 | 4 |
| | | (-4, 4) | (40, 40) | 45 | 9 |
| | | (-6, 6) | (10, 10) | 39 | 7 |
| | | (-8, 8) | (10, 10) | 42 | 9 |
| | 20 | (-4, 4) | (10, 10) | 10 | 1 |
| | 40 | (-4, 4) | (10, 10) | 21 | 1 |

(b) Experiments on MNIST and Fashion-MNIST

| Spline Order | Grids | Grid Range | Layers | Acc.@1 MNIST | Acc.@1 FMNIST |
|---|---|---|---|---|---|
| 1 | 10 | (-4, 4) | (10, 10) | 92 | 85 |
| 3 | 10 | (-4, 4) | (10, 10) | 94 | 86 |
| 4 | 10 | (-4, 4) | (10, 10) | 94 | 86 |
| 5 | 10 | (-1, 1) | (10, 10) | 89 | 83 |
| | | (-2, 2) | (10, 10) | 90 | 95 |
| | | (-4, 4) | (10, 10) | 95 | 86 |
| | | (-4, 4) | (20, 40) | 96 | 87 |
| | | (-4, 4) | (40, 40) | 96 | 88 |
| | | (-6, 6) | (10, 10) | 95 | 86 |
| | | (-8, 8) | (10, 10) | 95 | 86 |
| | 20 | (-4, 4) | (10, 10) | 93 | 85 |
| | 40 | (-4, 4) | (10, 10) | 91 | 83 |

computational requirement; a higher $r$ would incur a higher compute time as the size of the operator $\widehat{\Phi}$ scales with $r$.

### C.4.3 ENFORCING AN EXTREME LOW RANK STRUCTURE IN FOURIER KARAT

While our modular design of Fourier KArAt avoids the computational requirement of $\Phi^{i,j}$ by using a low hidden dimension $r$, it also enforces a low-rank structure in the attention matrix $\mathcal{A}^{i,j}$. In this context, we attempt to find the best possible value for $r$; see §C.4.2. However, we want to investigate how low the hidden dimension $r$ can be used without substantially compromising the performance of

Table 17: **Ablation on grid size** $G$ for ViT-Tiny, ViT-Small and ViT-Base. We find a particular grid size suitable for each of the models.

| Model | Acc.@1 | |
|---|---|---|
| | **CIFAR-10** | **CIFAR-100** |
| ViT-Base | 83.45 | 58.07 |
| $+ G_1 B$ | 81.81 | 55.92 |
| $+ G_1 U$ | 80.75 | 57.36 |
| $+ G_3 B$ | 80.09 | 56.01 |
| $+ G_3 U$ | 81.00 | 57.15 |
| $+ G_5 B$ | 79.80 | 54.83 |
| $+ G_5 U$ | 81.17 | 56.38 |
| $+ G_{11} B$ | 50.47 | 42.02 |
| $+ G_{11} U$ | 40.74 | 39.85 |
| ViT-Small | 81.08 | 53.47 |
| $+ G_1 B$ | 79.00 | 53.07 |
| $+ G_1 U$ | 66.18 | 53.86 |
| $+ G_3 B$ | 79.78 | 54.11 |
| $+ G_3 U$ | 79.52 | 53.86 |
| $+ G_5 B$ | 78.64 | 53.42 |
| $+ G_5 U$ | 78.75 | 54.21 |
| $+ G_{11} B$ | 77.39 | 52.62 |
| $+ G_{11} U$ | 78.57 | 53.35 |
| ViT-Tiny | 72.76 | 43.53 |
| $+ G_1 B$ | 75.75 | 45.77 |
| $+ G_1 U$ | 74.94 | 46.00 |
| $+ G_3 B$ | 76.69 | 46.29 |
| $+ G_3 U$ | 75.56 | 46.75 |
| $+ G_5 B$ | 75.85 | – |
| $+ G_5 U$ | 74.71 | – |
| $+ G_7 B$ | 75.11 | – |
| $+ G_7 U$ | 74.45 | – |
| $+ G_9 B$ | 74.85 | – |
| $+ G_9 U$ | 73.97 | – |
| $+ G_{11} B$ | 74.52 | – |
| $+ G_{11} U$ | 73.58 | – |

Table 18: **Ablation on hidden dimension** $r$ on CIFAR-10 with ViT-Base. Here, we compare the values of $r \in \{8, 12, 24\}$.

| Model | $r$ | Acc.@1 on CIFAR-10 |
|---|---|---|
| ViT-Base | – | 58.07 |
| $+ G_3 B$ | 24 | 80.54 |
| $+ G_3 U$ | 24 | 80.81 |
| $+ G_5 B$ | 24 | 77.99 |
| $+ G_5 U$ | 24 | 80.52 |
| $+ G_3 B$ | 12 | 80.09 |
| $+ G_3 U$ | 12 | 81.00 |
| $+ G_5 B$ | 12 | 79.80 |
| $+ G_5 U$ | 12 | 81.17 |
| $+ G_3 B$ | 8 | 80.76 |
| $+ G_3 U$ | 8 | 80.40 |
| $+ G_5 B$ | 8 | 79.79 |
| $+ G_5 U$ | 8 | 80.83 |

Table 19: Comparing performance of Fourier KArAt for ViT-Tiny and ViT-Base with or without using $\ell_1$ projection in Algorithm 1. ViT-Tiny and ViT-Base use traditional `softmax` and do not require $\ell_1$ projection.

| Model | $\ell_1$ Projection | Acc.@1 on CIFAR-10 |
|---|---|---|
| ViT- Tiny | ✗ | 72.76 |
| $+ G_3 B$ | ✓ | 41.99 |
| $+ G_3 U$ | ✓ | 40.85 |
| $+ G_3 B$ | ✗ | 76.69 |
| $+ G_3 U$ | ✗ | 75.56 |
| ViT-Base | ✗ | 83.45 |
| $+ G_3 B$ | ✓ | 47.44 |
| $+ G_3 U$ | ✓ | 46.11 |
| $+ G_3 B$ | ✗ | 80.09 |
| $+ G_3 U$ | ✗ | 81.00 |

learnable Fourier KArAt. The significance of this question lies in the information bottleneck created by an extremely low hidden dimension $r$ that helps to understand the tradeoff between computing requirements and final trained model quality. To this end, we experiment with extreme low-rankness in the hidden dimension, $r = 2, 4$, and report our findings in Table 20. Although $r = 2, 4$ are not optimal, and we observe an insignificant drop in the performance across both datasets (CIFAR-10 and CIFAR-100), it comes at a reduced computing requirement, especially required VRAM; see GPU memory in Table 20. For instance, ViT-Tiny+$G_3 B$ with $r = 2$ outperforms the vanilla variant, only has a modest 0.03M parameters increment from the vanilla ViT-Tiny, albeit similar GFLOPs, but an extra 1.07GB of GPU memory usage. Compared to the $r = 12$ variant, the memory usage is 3.47 GB lower, a total of 0.73M parameters less, but the relative decrease in accuracy is only 2.96% lower on the CIFAR-10 dataset. In ViT-Base backbones, the performance remains comparable, and even in the case of ViT-Base+$G_1 B$ on CIFAR-100, the variant with $r = 2$ surpasses the original variant with $r = 12$. Although this is not a concrete strategy, this observation supports our claim on probable research directions involving parameter reduction (discussed in §6) to improve scalability in such over-parameterized models. However, two such concrete strategies have been explored in §7, and we show that guided parameter reduction or attention sparsification can improve the scalability by large margins; see Table 5 and 6.

### C.4.4 ABLATION ON THE IMPACT OF GRID SIZE, $G$ ON FOURIER KARAT

KANs are highly dependent on certain hyperparameters, and Fourier-KArAt has only one hyperparameter to tune the performance — grid size $G$. Thus, we perform extensive experiments involving grid size $G$ and present them in Table 17. We observe that each of the particular ViT models, in conjunction with particular Fourier-KArAt variants, has a typical $G$ value that brings out its best performance, and there is no universal value of $G$ to follow. When performing validation with

Table 20: Complete compute requirement of ViT-Tiny+Fourier KArAt with extremely low hidden dimension training.

| Model | $r$ | Acc.@1 | | Parameters | GFLOPs | GPU Memory |
|-------|-----|--------|--|------------|--------|------------|
| | | CIFAR-10 | CIFAR-100 | | | |
| ViT-Tiny | | 72.76 | 43.53 | 5.53M | 1.262 | 2.94GB |
| $+ G_3 B$ | 12 | 76.69 | 46.29 | 6.29M | 1.425 | 7.48GB |
| | 4 | 75.90 | 45.76 | 5.80M | 1.313 | 4.68GB |
| | 2 | 74.39 | 44.36 | 5.56M | 1.285 | 4.01GB |
| $+ G_3 U$ | 12 | 75.56 | 46.75 | 5.59M | 1.425 | 7.29GB |
| | 4 | 73.13 | 44.98 | 5.57M | 1.313 | 5.08GB |
| | 2 | 71.56 | 42.43 | 5.55M | 1.285 | 4.41GB |
| ViT-Base | | 83.45 | 58.07 | 85.81M | 17.595 | 7.44GB |
| $+ G_1 B$ | 12 | 81.81 | 55.92 | 87.51M | 17.847 | 17.36GB |
| | 4 | 81.20 | 55.29 | 86.41M | 17.668 | 12.05GB |
| | 2 | 81.55 | 56.49 | 86.13M | 17.623 | 9.48GB |
| $+ G_1 U$ | 12 | 80.75 | 57.36 | 85.95M | 17.847 | 16.97GB |
| | 4 | 79.69 | 55.70 | 85.89M | 17.668 | 12.11GB |
| | 2 | 80.36 | 54.04 | 85.87M | 17.623 | 9.92GB |

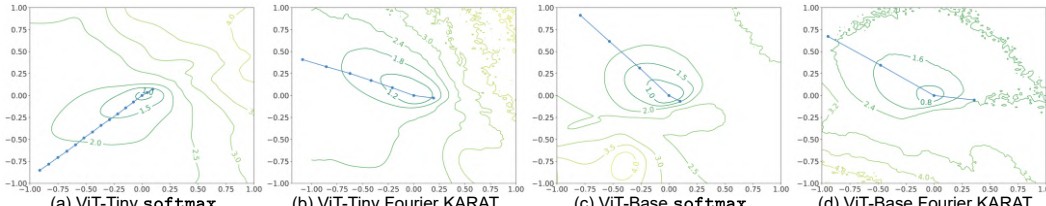

(a) ViT-Tiny `softmax`    (b) ViT-Tiny Fourier KARAT    (c) ViT-Base `softmax`    (d) ViT-Base Fourier KARAT

Figure 10: **Optimization path for ViT-Tiny and ViT-Base** (the smallest and the largest model) along the two largest principal component directions of the successive change of model parameters. We show the loss contours along with the trajectory of the optimizers. Perturbed contours indicate corresponding non-smooth, spiky loss surfaces.

ViT-Base+ variants on CIFAR-10 and CIFAR-100, the accuracy drops as the grid size passes a size after 5. However, this behavior is not persistent with the ViT-Small/Tiny+ variants; see Table 17[3].

### C.4.5    DOES FOURIER KARAT REQUIRE $\ell_1$-PROJECTION?

To ensure that each row vector of the learned attention matrix lies on a probability simplex, we project it onto the $\ell_1$-unit ball. We use Algorithm 1 to project learned attention vectors to the probability simplex and compare Top-1 accuracies to the baseline model. We note that using $\ell_1$ projection does not substantially increase the training time. However, from Table 19, we observe that incorporating this algorithm in the Fourier-KArAt does not improve its performance; instead, the performance significantly degrades. Algorithm 1 can be used with the choice of any basis function. However, we cannot comment on the role of $\ell_1$-projection in KArAt's performance with basis functions other than Fourier.

### C.4.6    MORE DISCUSSION ON BLOCKWISE AND UNIVERSAL CONFIGURATION

Although the primary inspiration behind the universal configuration comes from the shared basis functions in KAT Yang & Wang (2025), it aligns with the fact that `softmax` is a fixed function conventionally being used across all heads of all encoder layers in the ViT architectures. This raises the question of whether the token-to-token interaction in the self-attention can be modeled using a single universal function, or each instance of attention operations needs dynamic modeling using learnable functions. To investigate further, we compared the learned coefficients $\{a_{pqm}, b_{pqm}\}$ from Equation 2 and the weight matrix $W$ of the linear projector following the learnable basis, for each head across all the encoder layers in ViT-Tiny+Fourier KArAt. Thus, we obtain 36 sets (3 heads for

---

[3]We did not perform gridsize 5, 7, 9, and 11 experiments with ViT-Tiny+KArAt as it was already outperforming the base ViT-Tiny with gridsize 1 and 3, and the experiments are extensively resource intensive.

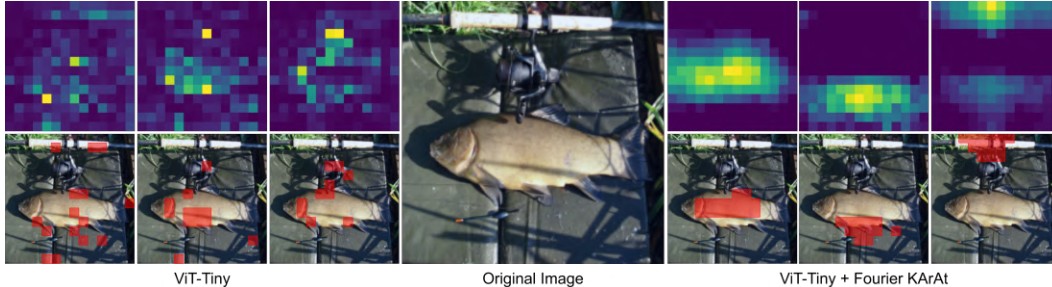

ViT-Tiny        Original Image        ViT-Tiny + Fourier KArAt

Figure 11: **Vit-Tiny attention map characterization.** Original image for inference (the center), the attention maps (top row), and contributing image regions (bottom row) for all three heads in ViT-Tiny: traditional MHSA (left) and Fourier KArAt $G_3B$ (right). The traditional MHSA sporadically focuses on fine-grained features of the primary object in the image. In contrast, the learnable attention in Fourier KArAt identifies the primary object features present significantly across all heads.

Table 21: **Ablation study on the operator variants** in Fourier KArAt. Note $\widehat{\Phi}(W\mathcal{A})$ and $W\widehat{\Phi}(\mathcal{A})$ refer to order of using linear projector $W$ and operator $\widehat{\Phi}$, and $\widehat{\Phi}_{[2]}(\widehat{\Phi}_{[1]}(\mathcal{A}))$ refers to two consecutive learnable operators $\widehat{\Phi}_{[1]}$ and $\widehat{\Phi}_{[2]}$.

| Model | Acc.@1 on CIFAR-10 | | |
|---|---|---|---|
| | $\widehat{\Phi}(W\mathcal{A})$ | $W\widehat{\Phi}(\mathcal{A})$ | $\widehat{\Phi}_{[2]}(\widehat{\Phi}_{[1]}(\mathcal{A}))$ |
| ViT-Base | | | |
| $+ G_{11}B$ | 41.02 | 50.47 | 30.14 |
| $+ G_{11}U$ | – | 47.74 | – |
| ViT-Tiny | | | |
| $+ G_{11}B$ | 67.04 | 75.41 | 33.09 |
| $+ G_{11}U$ | 70.98 | 73.67 | 36.98 |

Table 22: Performance using `softmax` and variants of Fourier KArAt in separate heads.

| Model | Low Rank Configurations (for Fourier KArAt) | No. of Distinct Heads | | Acc.@1 on CIFAR-10 |
|---|---|---|---|---|
| | | `softmax` | Fourier KArAt | |
| ViT-Base | | 12 | 0 | 83.45 |
| ViT-Base $+ G_{11}B$ | $\hat{\sigma} = \widehat{\Phi}(W\mathcal{A})$ | 0 | 12 | 41.02 |
| | $\hat{\sigma} = \widehat{\Phi}(W\mathcal{A})$ | 6 | 6 | 47.60 |
| | $\hat{\sigma} = W\widehat{\Phi}(\mathcal{A})$ | 0 | 12 | 50.47 |
| | $\hat{\sigma} = \widehat{\Phi}_{[2]}(\widehat{\Phi}_{[1]}(\mathcal{A}))$ | 0 | 12 | 30.14 |
| | $\hat{\sigma} = \widehat{\Phi}_{[2]}(\widehat{\Phi}_{[1]}(\mathcal{A}))$ | 6 | 6 | 30.24 |
| ViT-Tiny | | 3 | 0 | 72.76 |
| ViT-Tiny $+ G_{11}B$ | $\hat{\sigma} = \widehat{\Phi}_{[1]}(\widehat{\Phi}_{[2]}(\mathcal{A}))$ | 0 | 3 | 33.09 |
| | $\hat{\sigma} = \widehat{\Phi}_{[2]}(\widehat{\Phi}_{[1]}(\mathcal{A}))$ | 1 | 2 | 32.51 |
| | $\hat{\sigma} = \widehat{\Phi}_{[2]}(\widehat{\Phi}_{[1]}(\mathcal{A}))$ | 2 | 1 | 34.73 |
| ViT-Tiny $+ G_{11}U$ | $\hat{\sigma} = \widehat{\Phi}_{[2]}(\widehat{\Phi}_{[1]}(\mathcal{A}))$ | 0 | 3 | 36.98 |
| | $\hat{\sigma} = \widehat{\Phi}_{[2]}(\widehat{\Phi}_{[1]}(\mathcal{A}))$ | 1 | 2 | 33.12 |
| | $\hat{\sigma} = \widehat{\Phi}_{[2]}(\widehat{\Phi}_{[1]}(\mathcal{A}))$ | 2 | 1 | 70.57 |

each of the 12 layers) of such coefficients and weights from the model with blockwise configuration $G_3B$, and 3 sets from the model with universal configuration $G_3U$. Although mostly centered at zero, the distribution of each of these sets of parameters varies significantly from the others. The difference is mostly visible in the presence of long tails and the length of the tails. This characteristic is present not only in the blockwise configuration, but the three heads of the universal configuration also show a similar pattern.

## C.5 Alternate Variants of Fourier KArAt

In this section, we experiment with different attention variants on Fourier KArAt.

### C.5.1 Alternative Approaches to the Lower Rank Attention Structures

In the main paper, we approximated the effect of the operator, $\Phi^{i,j} \in \mathbb{R}^{N \times N}$ casting it as the product of two rank-$r$ operators—one operator with learnable activation, $\widehat{\Phi}$, and the other is the learnable the linear transformation matrix, $W$ such that $\Phi = \widehat{\Phi}W$. Here, we abuse the notations for simplicity. However, a natural question is how many different configurations are possible with $\widehat{\Phi}$ and $W$ such that they can approximate the effect of $\Phi$. Specifically, we use the following configurations — (i)[4] $W\widehat{\Phi}(\mathcal{A})$, where $\widehat{\Phi} \in \mathbb{R}^{r \times N}$ and $W \in \mathbb{R}^{N \times r}$, (ii) $\widehat{\Phi}(W\mathcal{A})$, where $W \in \mathbb{R}^{r \times N}$ and $\widehat{\Phi} \in \mathbb{R}^{N \times r}$, and (iii) $\widehat{\Phi}_{[2]}(\widehat{\Phi}_{[1]}(\mathcal{A}))$, where $\widehat{\Phi}_{[1]} \in \mathbb{R}^{r \times N}$ and $\widehat{\Phi}_{[2]} \in \mathbb{R}^{N \times r}$ are low-rank operators for learning activation functions. In the first configuration, $\widehat{\Phi}(\cdot)$ acts on each row of $\mathcal{A}$ to produce an $r$-dimensional vector, and then $W$ projects it back to an $N$-dimensional subspace. In the second configuration, $W$,

---
[4]This configuration was used in the main paper.

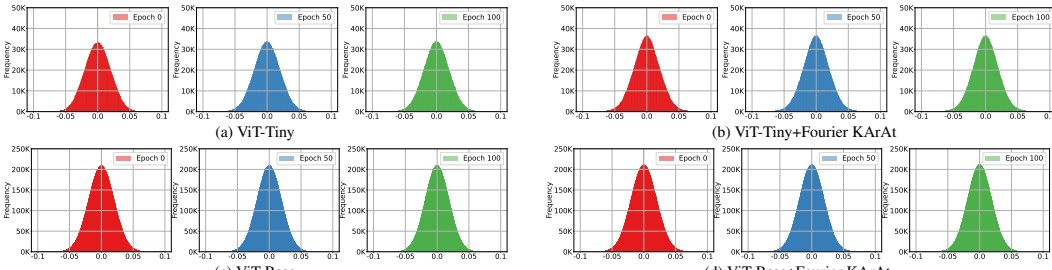

Figure 12: **Weight distribution** of ViT-Tiny and -Base with traditional MHSA and Fourier KArAt. The columns (left to right) represent weights at initialization, epoch 50, and epoch 100.

first projects each row of $\mathcal{A}$ to produce an $r$-dimensional vector, and then $\widehat{\Phi}$ with learnable activation produces $N$-dimensional vectors. In the third configuration, $\widehat{\Phi}_{[1]}$ first produces $r$-dimensional vector from rows of $\mathcal{A}$ by learning activations, and $\widehat{\Phi}_{[2]}$ obtains $N$-dimensional vectors by learning a second level of activations.

Primarily, we started with the full-rank operator $\Phi \in \mathbb{R}^{N \times N}$ and found its computations to be prohibitively expensive, regardless of the choice of the basis. Next, we conduct an ablation study to see which operator configuration works better; see Table 21. Our experiments show that the operator configuration (*ii*) demonstrates an inferior performance. We postulate that by down-projecting the $N$-dimensional attention row vector to a smaller dimension, $r$ loses adequate token-to-token interaction, and this fails to capture dependencies from significant attention units within a row. After that, from this limited information, learnable activation cannot significantly help the model's performance. On the other hand, (*iii*) also fails to perform due to its inability to model the attention well, despite having refined information, from the intermediate $r$-dimensional subspace. Overall, apart from configuration (*i*), the performances of the other configurations were inconsistent over multiple experiments. Also, they lag in training stability, particularly configuration (*iii*). Considering these observations, we proceeded to find the best possible grid size $G$ only with the configuration (*i*).

### C.5.2 FOURIER KARAT + SOFTMAX ATTENTION —A HYBRID VERSION

We consider another variant where we mix the learnable and pre-defined activation in each head. E.g., it is possible to have 2 of the 3 attention heads in each encoder block in ViT-Tiny activated by softmax and the third activated by the Fourier KArAt. With the idea of incorporating KAN to replace softmax activation, we are curious to see if the ViT model performs better with a hybrid mode of activation; see Table 22 for results.

### C.5.3 MORE COMBINATIONS

We have also experimented with more configurations of Fourier KArAt in various permutations of the strategies mentioned above and have not found any significant combination in terms of performance.

We also carefully design a particular training strategy where the attention operators ($\widehat{\Phi}$, $W$, and $\Phi$ wherever applicable) of Fourier KArAt are trained with a separate learning rate to alleviate the problem of the mismatch of the data passing through these learnable layers, as they consider each row of an attention matrix $\mathcal{A}$ as an input. The models have a gradient and loss explosion during training.

### C.6 DISTRIBUTION OF THE WEIGHTS

Zhang et al. (2022) considered an *invariant measure perspective* to describe the training loss stabilization of the neural networks. We adopt this idea to study the distribution of the weights of the smallest and largest models of the ViTs, Tiny and Base, and their Fourier KArAt variants. Figure 12 shows the distribution of the weights of these models during different training phases—The evolution of the weights' distributions for respective models remains invariant. Based on this observation, from Zhang et al. (2022), we can guarantee the convergence of loss values of all models; also, see Figure 6. However, as mentioned in Zhang et al. (2022), this perspective cannot comment on the generalization capacity and structural differences between different neural networks.

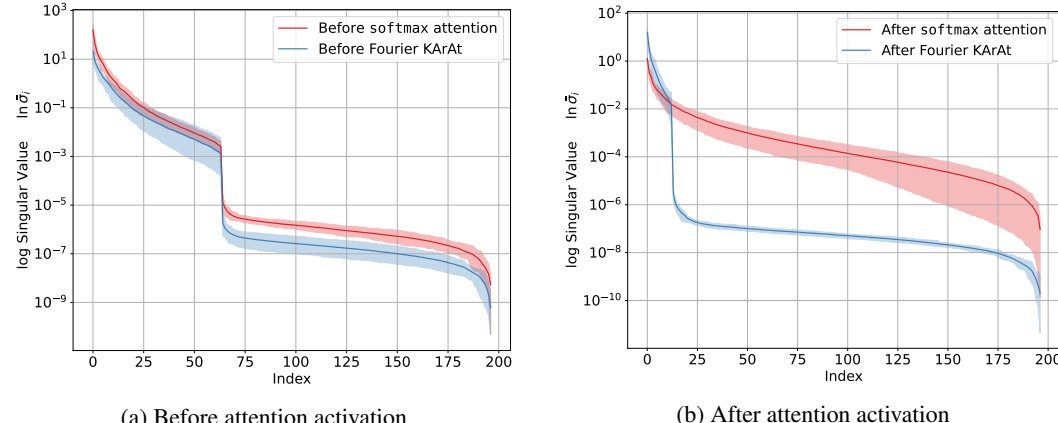

(a) Before attention activation

(b) After attention activation

Figure 13: **Spectral decomposition of the attention matrix** for ViT-Tiny on CIFAR-10 dataset with traditional `softmax` attention and our learnable Fourier KArAt. The traditional `softmax` attention and our learnable Fourier KArAt have almost similar low-rank structure, before activation functions are used.

## C.7 SPECTRAL ANALYSIS OF ATTENTION

Although we cannot comment on the generalizability by studying the distribution of the weights, from Table 1, we realize all the KArAt variants have more parameters than their vanilla counterparts. Therefore, it would be interesting to see how their attention matrices behave. As discussed in §4, the attention matrices in traditional MHSA have a low-rank structure. Following that study, we verified the apparent low-rank structure that Fourier KArAt's learned attention matrices possess.

We use all 3 heads in the last encoder block of ViT-Tiny on 5 randomly sampled images from the CIFAR-10 validation set. There are a total of 15 singular vectors (each of 197 dimensions) for any attention matrix of shape $197 \times 197$, where the singular values are arranged in non-increasing order. Let $\sigma_i$ be the $i^{\text{th}}$ singular value across all heads and samples (it is permutation invariant). For each $i \in [197]$, we plot $[\ln(\sigma_i^{\min}), \ln(\bar{\sigma}_i), \ln(\sigma_i^{\max})]$, where $\bar{\sigma}_i$ is the average of $i^{\text{th}}$-indexed singular value across all samples and heads.

To investigate the inherent low-rank structure of attention, we plot the natural logarithm of singular values of attention matrices, $\sigma_i$, before and after attention activation in Figure 13. From Figure 13a, we observe that the traditional `softmax` attention and our learnable Fourier KArAt have almost similar low-rank structures. However, Figure 13b shows that the traditional MHSA has significantly larger singular values than its KArAt variant, $G_3B$. It can also be noticed that before the activation, both traditional MHSA and Fourier KArAt feature a sharp drop in singular values between the 50th and 75th indices. This sharp drop in singular values vanishes in `softmax` attention, indicating a normalization. However, due to the hidden dimension $r$, Fourier KArAt enforces a much lower rank than the traditional MHSA.

Additionally, we observe from the weight distributions in Figure 12 that Fourier KArAt variants have more entries close to zero than their ViT counterparts. This, along with the low-rankness, can inspire low-rank + sparse training strategies of Fourier KArAt.

