# OpenReview forum: "Kolmogorov-Arnold Attention: Is Learnable Attention Better For Vision Transformers?"
_ICLR.cc/2026/Conference — Submitted to ICLR 2026_

### Official Review · Reviewer_eokU · 2025-10-29

**Soundness:** 2
**Presentation:** 3
**Contribution:** 2
**Rating:** 2
**Confidence:** 4

**Summary:**

This work investigates whether substituting the conventional softmax operation in ViTs with a learnable activation function will improve the performance or not. The proposed method draws inspiration from the new KAN architectures. A key implementation challenge is how to mitigate the computational cost when replacing it with a learnable activation function. The authors addressed this with a low-rank approximation strategy, where the learnable function could project the input to a lower-dimensional space. To improve the performance of the low-rank approximated method, the authors have also proposed using the Fourier basis encodings to replace the conventional B-splines. The authors have provided performance of the proposed method on ViT structures with various datasets, along with the intuitive analysis of the performance, and the qualitative result to show how the learnable activation is helping during the attention.

**Strengths:**

- The proposed method further explored the potential of using KAN-based structure in the widely used ViTs, which might make a good contribution to the vision community to push forward with the architecture update and exploration. In fact, to make the ViTs more “explainable” is an interesting and important research topic.
- The authors have made relatively thorough and rigorous experiments on various benchmarks with different model settings, including parameter comparison, ablation studies, and efficiency analysis, though some of the analysis could be moved to the main paper.

**Weaknesses:**

- I understand that making the activation function in transformers learnable and making it a KAN-based architecture is interesting and could have potential benefits to allow the attention mechanism to better learn the complex data. However, based on what has been shown in the main paper, the proposed architecture cannot be applied to larger-scale models and is limited to ViT-Tiny. I think this is a big disadvantage, and I do think having real-world applications on those larger-scale---or not even larger-scale, but rather “conventional scale” we would use in modern days---is very important for the general motivation of this paper. I do think the authors need to revisit the proposed method and reassure that this big discrepancy in performance on larger-scale models is rooted in the limitations of the KAN-based architecture or not. On the other hand, if the authors really want to focus only on the ViT-Tiny architecture, it will change the entire story of this paper. The motivation, the focus, and the theoretical explanation need to be updated.
- I think the loss landscape and the attention visualization should have provided more intuitions on why the proposed method is more explainable than the conventional self-attention, and the authors should have provided more theoretical links to the intuitive results. However, the main analysis in the paper is not presented theoretically or intuitively strong. I hope the authors could revise the paper and update it with a stronger analysis.

**Questions:**

I think the authors should focus first on the major weaknesses of the paper and truly update the manuscript based on reviewers’ suggestions.

---

> ### Author Response · Authors · 2025-11-26
> **Rebuttal**
>
> ### 1. **Scalability of KArAt and Performance in Larger ViTs**
>
> The fundamental challenge of training KArAt via the modern deep learning toolkits in the present-day GPU and CUDA interface remains its scalability with larger overparameterized models. Although the system-level modification is beyond the scope of this paper, we discuss two potential solutions that can alleviate the curse of overparameterization in KArAt: *(i)* **parameter reduction by using KArAt only in a few layers**, and *(ii)* **improving the training instability by attention sparsification** in **Section 7 Improving KArAt’s Scalability**. We show that both the strategies improve scalability for ViT-Small and -Base, and in some cases improve performance in ViT-Tiny as well. Using these two strategies, for ViT-Tiny, -Small, -Base, we show maximums of *7.89%, 3.89%, 4.16% improvements on CIFAR-10, and 16.10%, 12.55%, 10.50% improvement on CIFAR-100, respectively, over traditional softmax attention.* **On ImageNet-1K, the accuracy drop remains less than a modest 0.78% and 0.77% on the ViT-Base and -Small respectively; see Table 5. This is a vast improvement from the results in Table 1 where Fourier KArAt was showing a performance drop of 6.68% compared to ViT-Base, and 3.89% compared to ViT-Small.**
>
> Additionally, in Table 6, we show that the strategy *(i)* with Fourier KArAt outperforms the traditional MHSA on SVHN for the ViT-Base and ViT-Small, on Flowers 102 for ViT-Small and Vit-Tiny, and on STL-10 for ViT-Tiny. These empirical results prove that properly guided strategies can improve scalability of KArAt in larger models; see our discussion in Section 7 of the recently updated version of the paper.
>
> Table 5 is given below for the convenience of the reviewer:
>
> | Model | CIFAR-10 Acc.@1 | CIFAR-10 Acc.@5 | CIFAR-100 Acc.@1 | CIFAR-100 Acc.@5 | ImageNet-1K Acc.@1 | ImageNet-1K Acc.@5 | Parameters |
> | :--- | :--- | :--- | :--- | :--- | :--- | :--- | :--- |
> | **ViT-Base** | 83.45 | 99.19 | 58.07 | 83.70 | **72.90** | 90.56 | 85.81M |
> | + $[G_3B]_{12}$ | 86.74 (3.94%↑) | 99.51 | 64.07 (10.33%↑) | 87.29 | 72.33 (0.78%↓) | 89.99 | 86.82M (1.18%↑) |
> | + $G_3B^\dagger$ | **86.92 (4.16%↑)** | 99.42 | 63.49 (9.33%↑) | 88.14 | - | - | 88.87M (3.57%↑) |
> | + $G_3U^\dagger$ | 84.82 (1.64%↑) | 99.24 | **64.17 (10.50%↑)** | 87.93 | - | - | 86.06M (0.29%↑) |
> | **ViT-Small** | 81.08 | 99.02 | 53.47 | 82.52 | **70.50** | 89.34 | 22.05M |
> | + $[G_3B]_{12}$ | 83.86 (3.43%↑) | 99.39 | 60.16 (12.51%↑) | 86.44 | 69.96 (0.77%↓) | 88.89 | 22.18M (0.56%↑) |
> | + $G_3B^\dagger$ | **84.23 (3.89%↑)** | 99.30 | 59.27 (10.85%↑) | 85.87 | - | - | 23.58M (6.94%↑) |
> | + $G_3U^\dagger$ | 84.06 (3.68%↑) | 99.19 | **60.18 (12.55%↑)** | 86.30 | - | - | 22.18M (0.56%↑) |
> | **ViT-Tiny** | 72.76 | 98.14 | 43.53 | 75.00 | **59.15** | 82.07 | 5.53M |
> | + $[G_3B]_{12}$ | 77.93 (7.11%↑) | 98.76 | 50.54 (16.10%↑) | 80.54 | 58.98 (0.29%↓) | 81.91 | 5.59M (1.08%↑) |
> | + $G_3B^\dagger$ | 77.81 (6.94%↑) | 98.76 | 48.33 (11.03%↑) | 78.68 | - | - | 6.29M (13.74%↑) |
> | + $G_3U^\dagger$ | **79.08 (8.69%↑)** | 98.91 | **49.33 (13.32%↑)** | 79.40 | - | - | 5.59M (1.08%↑) |
>
> Table 6 is also given here for the convenience of the reviewer:
>
> | Model | SVHN (Acc.@1) | Flowers 102 (Acc.@1) | STL-10 (Acc.@1) |
> | :--- | :--- | :--- | :--- |
> | **ViT-Base** | 97.74 | **92.24** | **97.26** |
> | + $[G_3B]_{12}$ | **97.81** | 91.97 | 96.65 |
> | **ViT-Small** | 97.48 | 91.46 | **96.09** |
> | + $[G_3B]_{12}$ | **97.56** | **91.59** | 95.79 |
> | **ViT-Tiny** | **96.69** | 84.21 | 93.20 |
> | + $[G_3B]_{12}$ | 96.40 | **85.53** | **93.35** |

---

> > ### Author Response · Authors · 2025-11-26
> > **Rebuttal --- continued**
> >
> > ### 2. **Attention Visualization & Explainability**
> >
> > Self-attention is the heart of the transformer architectures that model token-to-token interactions and determine how such interactions would result in feature representations. Irrespective of tasks or datasets at hand, traditional MHSA uses softmax non-linearity as a **“one size fits all"** mechanism for token interactions without the necessary justification of using it. In this work, we introduce KArAt as an improved and interpretable architecture, which has more freedom in modeling the token interactions via learnable activation that depends on the choice of user-inferred basis. By introducing learnability in self-attention, we observe this interpretability directly reflected in KArAt’s attention heatmaps [1] visualization. We note that self-supervised models learn rich meaningful features and deep, robust representations [2], and usually attend to the entire object compared to spurious cues in supervised vanilla ViTs [3]. Chefer et al. [3] attends the entire object in their attention like self-supervision for improved interpretability and explainability. Extending this argument, we show that KArAt’s attention visualizations attend to the entire object and offer improved explainability; see our discussion on Lines 411-417.
> >
> > The study on attention transfer by Li et al. shows that the attention matrices are the most important part of the transformer and can bring improved performance over the vanilla supervised ViT; see Table 1 and Figure 1 in [4]. In the same spirit, we have trained vanilla ViTs with precomputed attention matrices (including the interaction scores responsible for attention maps) from its corresponding Fourier KArAt variants. Our experiments show that KArAt attention matrices do not just produce better visuals but also are directly responsible for the performance gains; see our discussion in Lines 417-434 and results in Table 4. We provide the results therein for the reviewer's convenience.
> >
> > | Base Model | Attention Matrix copied from | Acc.@1 on CIFAR-10 | Acc.@5 on CIFAR-10 | Acc.@1 on CIFAR-100 | Acc.@5 on CIFAR-100 |
> > | :--- | :--- | :--- | :--- | :--- | :--- |
> > | ViT-Tiny | None | 72.76 | 98.14 | 43.53 | 75.00 |
> > | ViT-Tiny+$G_3B$ | None | 76.69 | 98.57 | 46.29 | 77.02 |
> > | **ViT-Tiny** | **ViT-Tiny+$G_3B$** | **77.94** | **98.71** | **48.91** | **78.42** |
> > | ViT-Base | None | 83.45 | 99.19 | 58.07 | 83.70 |
> > | ViT-Base+$G_1B$ | None | 81.81 | 99.01 | 55.92 | 82.04 |
> > | **ViT-Base** | **ViT-Base+$G_1B$** | **81.34** | **98.70** | **55.33** | **80.80** |
> >
> > [1] Caron et al. "Emerging Properties in Self-Supervised Vision Transformers." ICCV, pp. 9650–9660, 2021.
> >
> > [2] Ericsson et al. "How well do self-supervised models transfer?" CVPR, pp. 5414-5423. 2021.
> >
> > [3] Chefer et al. "Optimizing relevance maps of vision transformers improves robustness." NeurIPS, 35, pp. 33618-33632. 2022.
> >
> > [4] Li et al. "On the surprising effectiveness of attention transfer for vision transformers." NeurIPS, 37, pp. 113963-113990. 2024.

---

### Official Review · Reviewer_aECS · 2025-10-30

**Soundness:** 3
**Presentation:** 3
**Contribution:** 2
**Rating:** 4
**Confidence:** 5

**Summary:**

This paper introduces Kolmogorov-Arnold Attention (KArAt), a learnable attention mechanism inspired by Kolmogorov-Arnold Networks (KANs). Unlike standard Transformers that use a fixed softmax operation to compute attention weights, KArAt replaces it with learnable nonlinear bases such as Fourier, B-spline, wavelet, and rational functions. To alleviate the huge memory overhead, the authors employ a low-rank approximation and evaluate Fourier-KArAt across multiple Vision Transformer (ViT) variants and datasets (CIFAR-10/100, ImageNet-1K). Results show noticeable gains on small models (ViT-Tiny) but consistent degradation on larger models and transfer learning tasks.

**Strengths:**

KArAt represents one of the first systematic attempts to reparameterize attention as a learnable functional operator, rather than a fixed probabilistic normalization. This perspective is intellectually stimulating and could influence future research in both theoretical deep learning and architectural design.

Another strong aspect is the experimental comprehensiveness.
The authors do not merely demonstrate their method. They conduct cross-scale evaluations (Tiny → Base), transfer learning tests, and multiple ablation studies on basis functions.

The writing and presentation quality are outstanding. The figures are informative, the mathematical notations are consistent, and the discussion sections are honest about both strengths and limitations.

**Weaknesses:**

While conceptually elegant, the proposed approach does not deliver consistent empirical gains. Performance drops substantially on larger ViT models, revealing poor scalability and unstable optimization dynamics. Although the learnable operator Φ enriches the model’s nonlinearity, it also sharpens the loss landscape, leading to unstable gradients and weaker generalization.

Moreover, the transfer learning results (Table 2) show clear degradation across datasets. Indicating that the method’s performance is also extremely sensitive to hyper-parameters and whether the operator is blockwise or universal. This sensitivity further limits the model’s practicality and reproducibility.

Finally, the paper does not clearly define the intended application domain—it remains unclear when or why one should prefer KArAt over standard or efficient attention variants (e.g., Performer [1], Linformer [2], cosFormer [3]).

[1] Choromanski, K.M., Likhosherstov, V., Dohan, D., Song, X., Gane, A., Sarlos, T., Hawkins, P., Davis, J.Q., Mohiuddin, A., Kaiser, L. and Belanger, D.B., Rethinking Attention with Performers. In International Conference on Learning Representations.
[2] Wang, S., Li, B.Z., Khabsa, M., Fang, H. and Ma, H., 2020. Linformer: Self-attention with linear complexity. arXiv preprint arXiv:2006.04768.
[3] Qin, Z., Sun, W., Deng, H., Li, D., Wei, Y., Lv, B., Yan, J., Kong, L. and Zhong, Y., cosFormer: Rethinking Softmax In Attention. In International Conference on Learning Representations.

**Questions:**

1. The authors attribute performance drops in larger ViTs to “spiky loss landscapes.”
Have they empirically verified this claim or could other regularization techniques help stabilize training?

2. Since many recent attention variants also modify the attention for efficiency or stability, how does KArAt compare in both computational cost and performance under comparable parameter budgets?

3. Can the authors identify specific domains where KArAt might offer concrete advantages over standard softmax?

4. Given that the Q, K, and V projections are already learnable and flexible, what motivates the replacement of the softmax operator with another learnable function? Could the authors provide concrete evidence that this additional nonlinearity offers expressive benefits beyond what the QKV transformations already achieve?

---

> ### Author Response · Authors · 2025-11-26
> **Rebuttal**
>
> ### 1. **Empirical Validation of Performance Degradation in Larger ViTs**
>
> In our analyses in Section 5.2, we dissect the model’s performance using the famous tools (a) loss landscape, (b) attention visualization, and we additionally discuss on (c) weight distribution and (d) spectral analysis in Sections C.6 and C.7, respectively. Out of these four tools, the first two strongly validate the better performance of KArAt in smaller ViTs, and the performance degradation of KArAt in larger ViTs. The loss landscape analysis shows that the loss surface of ViT-Base+Fourier KArAt is most spiky, making it a narrow margin model with sharp minima in which small perturbations in the parameter space lead to high misclassification due to their exponentially larger volume in high-dimensional spaces. This also indicates that the model is overparameterized and prone to training instability. These are well known facts supported by the seminal papers [1, 2]. We used these results to showcase the performance degradation of Fourier KArAt variants in larger ViTs and postulate that reducing the number of learnable parameters in KArAt could lead to better performance in larger ViTs.
>
> To empirically validate these, we employ two strategies --- *(i)* **parameter reduction by using KArAt only in a few layers**, and *(ii)* **improving the training instability by attention sparsification** in **Section 7 Improving KArAt’s Scalability**. We show that both the strategies improve scalability for ViT-Small and -Base, and in some cases improve performance in ViT-Tiny as well. Using these two strategies, for ViT-Tiny, -Small, -Base, we show maximums of *7.89%, 3.89%, 4.16% improvements on CIFAR-10, and 16.10%, 12.55%, 10.50% improvement on CIFAR-100, respectively, over traditional softmax attention.* **On ImageNet-1K, the accuracy drop remains less than a modest 0.78% and 0.77% on the ViT-Base and -Small respectively; see Table 5. This is a vast improvement from the results in Table 1 where Fourier KArAt was showing a performance drop of 6.68% compared to ViT-Base, and 3.89% compared to ViT-Small.**
>
> Additionally, in Table 6, we show that the strategy *(i)* with Fourier KArAt outperforms the traditional MHSA on SVHN for the ViT-Base and ViT-Small, on Flowers 102 for ViT-Small and Vit-Tiny, and on STL-10 for ViT-Tiny. These empirical results prove that properly guided strategies can improve scalability of KArAt in larger models; see our discussion in Section 7 of the recently updated version of the paper.
>
> Table 5 is given below for the convenience of the reviewer:
>
> | Model | CIFAR-10 Acc.@1 | CIFAR-10 Acc.@5 | CIFAR-100 Acc.@1 | CIFAR-100 Acc.@5 | ImageNet-1K Acc.@1 | ImageNet-1K Acc.@5 | Parameters |
> | :--- | :--- | :--- | :--- | :--- | :--- | :--- | :--- |
> | **ViT-Base** | 83.45 | 99.19 | 58.07 | 83.70 | **72.90** | 90.56 | 85.81M |
> | + $[G_3B]_{12}$ | 86.74 (3.94%↑) | 99.51 | 64.07 (10.33%↑) | 87.29 | 72.33 (0.78%↓) | 89.99 | 86.82M (1.18%↑) |
> | + $G_3B^\dagger$ | **86.92 (4.16%↑)** | 99.42 | 63.49 (9.33%↑) | 88.14 | - | - | 88.87M (3.57%↑) |
> | + $G_3U^\dagger$ | 84.82 (1.64%↑) | 99.24 | **64.17 (10.50%↑)** | 87.93 | - | - | 86.06M (0.29%↑) |
> | **ViT-Small** | 81.08 | 99.02 | 53.47 | 82.52 | **70.50** | 89.34 | 22.05M |
> | + $[G_3B]_{12}$ | 83.86 (3.43%↑) | 99.39 | 60.16 (12.51%↑) | 86.44 | 69.96 (0.77%↓) | 88.89 | 22.18M (0.56%↑) |
> | + $G_3B^\dagger$ | **84.23 (3.89%↑)** | 99.30 | 59.27 (10.85%↑) | 85.87 | - | - | 23.58M (6.94%↑) |
> | + $G_3U^\dagger$ | 84.06 (3.68%↑) | 99.19 | **60.18 (12.55%↑)** | 86.30 | - | - | 22.18M (0.56%↑) |
> | **ViT-Tiny** | 72.76 | 98.14 | 43.53 | 75.00 | **59.15** | 82.07 | 5.53M |
> | + $[G_3B]_{12}$ | 77.93 (7.11%↑) | 98.76 | 50.54 (16.10%↑) | 80.54 | 58.98 (0.29%↓) | 81.91 | 5.59M (1.08%↑) |
> | + $G_3B^\dagger$ | 77.81 (6.94%↑) | 98.76 | 48.33 (11.03%↑) | 78.68 | - | - | 6.29M (13.74%↑) |
> | + $G_3U^\dagger$ | **79.08 (8.69%↑)** | 98.91 | **49.33 (13.32%↑)** | 79.40 | - | - | 5.59M (1.08%↑) |
>
> Table 6 is also given here for the convenience of the reviewer:
>
> | Model | SVHN (Acc.@1) | Flowers 102 (Acc.@1) | STL-10 (Acc.@1) |
> | :--- | :--- | :--- | :--- |
> | **ViT-Base** | 97.74 | **92.24** | **97.26** |
> | + $[G_3B]_{12}$ | **97.81** | 91.97 | 96.65 |
> | **ViT-Small** | 97.48 | 91.46 | **96.09** |
> | + $[G_3B]_{12}$ | **97.56** | **91.59** | 95.79 |
> | **ViT-Tiny** | **96.69** | 84.21 | 93.20 |
> | + $[G_3B]_{12}$ | 96.40 | **85.53** | **93.35** |

---

> > ### Author Response · Authors · 2025-11-26
> > **Rebuttal --- continued**
> >
> > ### 2. **Efficient Attentions vs. KArAt**
> >
> > We are aware of the efficient attentions that use linear/kernelized attentions for efficiency and consider them orthogonal to the direction of introducing learnable components for improved modeling of the relationship among the token interactions. We acknowledge that KArAt increases the number of parameters, FLOPs, GPU memory requirements, and training time (Tables 1 and 15, and Figure 7) for the vision transformers. However, *the goal of this work is not to produce more efficient or stable attention, and we explicitly mention this in the main paper*; see **Lines 99-105 in the Introduction.** Instead, **KArAt is the first attempt to introducing learnability in self-attention in ViTs** that offers added flexibility and adaptability in modeling token interactions. We discuss KArAt’s improved interpretability and explainability in Section 5.3. The performance gain can be observed in Tables 5 and 6 where we address the scalability problem and reduce the effective trainable parameters. This was our attempt to make the learnable attention efficient.
> >
> > ### 3. **KArAt over QKV interactions and softmax**
> >
> > Stemming from Kolmogorov-Arnold Representation Theorem (Theorem 1 in our paper, proposed by Kolmogorov and Arnold in circa 1956), KANs with learnable activation functions can approximate complex functions and provide an interpretable alternative to MLPs [3, 4, 5] and can facilitate meaningful interaction between the model and human intuition. In the same spirit, Theorem 2.1 in the original KAN paper states that if the target function admits a structure of the composition of $L$-KAN layers with smooth activation functions, then we can use an $L$-layer KAN (with the choice of B-Spline basis function) to approximate it well. This allows them to model complex relations that lie in the data distribution. The authors of KAN claimed that KANs can discover the need for a new function whose numerical behavior suggests it may be a Bessel function; see Figure 23 (d) in [3]. In fact, Figure 4(e) in [3] shows an example where KANs are very close to discovering a new equation.
> >
> > While there is no direct connection between KANs and MHSA, by using a similar argument, we hypothesize that KANs could be a better fit for approximating the underlying non-linear relationship among the interactions between image patches and can be used for modeling token interactions with learnable components. Hence, KArAt can be viewed as an upgrade to vanilla self-attention, offering adaptability and more freedom for modeling token interactions in an interpretable way. KArAt provides the freedom to choose from a multitude of simple, easy-to-interpret, and tunable functions whose linear combination can produce a large class of unknown functions, rather than a fixed and known function, softmax. The traditional softmax attention adds a non-linearity for modeling such interactions in a naive *“one-size-fits-all"* mechanism without any justification. This interpretability is directly reflected in KArAt’s attention maps; see Figures 4 and 11. Moreover, the benefits of learnability in KArAt are also directly reflected in the performance improvement shown in Tables 5 and 6; see our detailed discussion on performance gain in Section 7, Lines 520-531 of the updated paper.
> >
> > [1] Hao Li, Zheng Xu, Gavin Taylor, Christoph Studer, and Tom Goldstein. Visualizing the Loss Landscape of Neural Nets. NeurIPS, 31, 2018.
> >
> > [2] Ronny Huang, Zeyad Emam, Micah Goldblum, Liam Fowl, Justin K. Terry, Furong Huang, and Tom Goldstein. Understanding Generalization Through Visualizations. "I Can’t Believe It’s Not Better!" at NeurIPS Workshops, 137, pp. 87–97, 2020.
> >
> > [3] Liu et al. "KAN: Kolmogorov-Arnold Networks." ICLR, 2025.
> >
> > [4] Liu et al. "KAN 2.0: Kolmogorov-Arnold Networks Meet Science." Preprint arXiv:2408.10205, 2024.
> >
> > [5] Yang and Wang. "Kolmogorov-Arnold Transformer." ICLR, 2025.

---

### Official Review · Reviewer_ouZw · 2025-11-01

**Soundness:** 3
**Presentation:** 3
**Contribution:** 2
**Rating:** 4
**Confidence:** 3

**Summary:**

The paper proposes a learnable attention mechanism called KArAt, claimed to be the first learnable attention mechanism that replaces the traditional softmax function. KArAt uses learnable activation functions based on Fourier, Wavelets and B-splines. The paper uses a low-rank approximation to overcome the memory constraints. The authors' experiment on standard vision datasets like CIFAR-10, ImageNet-1K shows 5-7% improvements over the baseline, but fails to adapt to larger ViTs, showing poorer performance than the baselines. The method is also more computationally expensive than the baseline, requiring more memory and training time. The authors claim with extensive analysis that the KArAt produces better attention maps and transfers well to various downstream tasks.

**Strengths:**

1) First of its kind analysis of learnable attention on ViT transformers through thorough analysis and ablation on various model sizes, various downstream tasks visualizing the loss landscape, attention map and spectral analysis. The authors are honest with the flaws in the approach (doesn’t scale well and is computationally expensive).
2) KArAt produces better attention maps when compared to the traditional MHSA.

**Weaknesses:**

1) Although the authors claim KArAt to produce better attention maps through attention heatmap and spectral analysis, there is no numerical evidence that shows this result. Maybe curating a small dataset derived from an existing one that doesn’t work well with MHSA but with KArAt would have been great addition.
2) Gains concentrate on Tiny; Small/Base underperform MHSA, and training costs are significantly higher. If KArAt doesn’t stabilize for larger backbones, the usability case is weak.

**Questions:**

1) What aspects of KArAt most directly cause degradation on Small/Base? Any signs that per‑layer/ per‑head G or r can stabilize training at scale?
2) How does having a few KArAt layers mixed with MHSA affect accuracy and stability on ViT-Tiny/Small/Base? Does placement (early/mid/late) matter under compute-matched training

---

> ### Author Response · Authors · 2025-11-26
> **Rebuttal**
>
> ### 1. **On KArAt’s Better Attention Beyond Attention Map Visualization**
>
> We have used DINO [1] styled attention heatmaps for visualization. We note that self-supervised models learn rich meaningful features and deep, robust representations [2], and usually attend to the entire object compared to spurious cues in supervised vanilla ViTs [3]. Chefer et al. [3] attends the entire object in their attention like self-supervision for improved interpretability and explainability. Extending this argument, we show that KArAt’s attention visualizations attend to the entire object and offer improved explainability; see our discussion on Lines 411-417.
>
> The study on attention transfer by Li et al. shows that the attention matrices are the most important part of the transformer and can bring improved performance over the vanilla supervised ViT; see Table 1 and Figure 1 in [4]. In the same spirit, we have trained vanilla ViTs with precomputed attention matrices (including the interaction scores responsible for attention maps) from its corresponding Fourier KArAt variants. Our experiments show that KArAt attention matrices do not just produce better visuals but also are directly responsible for the performance gains; see our discussion in Lines 417-434 and results in Table 4. For the reviewer’s convenience, we provide the results below:
>
> | Base Model | Attention Matrix copied from | Acc.@1 on CIFAR-10 | Acc.@5 on CIFAR-10 | Acc.@1 on CIFAR-100 | Acc.@5 on CIFAR-100 |
> | :--- | :--- | :--- | :--- | :--- | :--- |
> | ViT-Tiny | None | 72.76 | 98.14 | 43.53 | 75.00 |
> | ViT-Tiny+$G_3B$ | None | 76.69 | 98.57 | 46.29 | 77.02 |
> | **ViT-Tiny** | **ViT-Tiny+$G_3B$** | **77.94** | **98.71** | **48.91** | **78.42** |
> | ViT-Base | None | 83.45 | 99.19 | 58.07 | 83.70 |
> | ViT-Base+$G_1B$ | None | 81.81 | 99.01 | 55.92 | 82.04 |
> | **ViT-Base** | **ViT-Base+$G_1B$** | **81.34** | **98.70** | **55.33** | **80.80** |
>
> [1] Caron et al. "Emerging Properties in Self-Supervised Vision Transformers." ICCV, pp. 9650–9660, 2021.
>
> [2] Ericsson et al. "How well do self-supervised models transfer?" CVPR, pp. 5414-5423. 2021.
>
> [3] Chefer et al. "Optimizing relevance maps of vision transformers improves robustness." NeurIPS, 35, pp. 33618-33632. 2022.
>
> [4] Li et al. "On the surprising effectiveness of attention transfer for vision transformers." NeurIPS, 37, pp. 113963-113990. 2024.

---

> ### Author Response · Authors · 2025-11-26
> **Rebuttal --- continued**
>
> ### 2. **Scalability and Performance of KArAt**
>
> The fundamental challenge of training KArAt via the modern deep learning toolkits in the present-day GPU and CUDA interface remains its scalability with larger overparameterized models. Although the system-level modification is beyond the scope of this paper, we discuss two potential solutions that can alleviate the curse of overparameterization in KArAt: *(i)* **parameter reduction by using KArAt only in a few layers**, and *(ii)* **improving the training instability by attention sparsification** in **Section 7 Improving KArAt’s Scalability**. We show that both the strategies improve scalability for ViT-Small and -Base, and in some cases improve performance in ViT-Tiny as well. Using these two strategies, for ViT-Tiny, -Small, -Base, we show maximums of *7.89%, 3.89%, 4.16% improvements on CIFAR-10, and 16.10%, 12.55%, 10.50% improvement on CIFAR-100, respectively, over traditional softmax attention.* **On ImageNet-1K, the accuracy drop remains less than a modest 0.78% and 0.77% on the ViT-Base and -Small respectively; see Table 5. This is a vast improvement from the results in Table 1 where Fourier KArAt was showing a performance drop of 6.68% compared to ViT-Base, and 3.89% compared to ViT-Small.**
>
> Additionally, in Table 6, we show that the strategy *(i)* with Fourier KArAt outperforms the traditional MHSA on SVHN for the ViT-Base and ViT-Small, on Flowers 102 for ViT-Small and Vit-Tiny, and on STL-10 for ViT-Tiny. These empirical results prove that properly guided strategies can improve scalability of KArAt in larger models; see our discussion in Section 7 of the recently updated version of the paper.
>
> Table 5 is given below for the convenience of the reviewer:
>
> | Model | CIFAR-10 Acc.@1 | CIFAR-10 Acc.@5 | CIFAR-100 Acc.@1 | CIFAR-100 Acc.@5 | ImageNet-1K Acc.@1 | ImageNet-1K Acc.@5 | Parameters |
> | :--- | :--- | :--- | :--- | :--- | :--- | :--- | :--- |
> | **ViT-Base** | 83.45 | 99.19 | 58.07 | 83.70 | **72.90** | 90.56 | 85.81M |
> | + $[G_3B]_{12}$ | 86.74 (3.94%↑) | 99.51 | 64.07 (10.33%↑) | 87.29 | 72.33 (0.78%↓) | 89.99 | 86.82M (1.18%↑) |
> | + $G_3B^\dagger$ | **86.92 (4.16%↑)** | 99.42 | 63.49 (9.33%↑) | 88.14 | - | - | 88.87M (3.57%↑) |
> | + $G_3U^\dagger$ | 84.82 (1.64%↑) | 99.24 | **64.17 (10.50%↑)** | 87.93 | - | - | 86.06M (0.29%↑) |
> | **ViT-Small** | 81.08 | 99.02 | 53.47 | 82.52 | **70.50** | 89.34 | 22.05M |
> | + $[G_3B]_{12}$ | 83.86 (3.43%↑) | 99.39 | 60.16 (12.51%↑) | 86.44 | 69.96 (0.77%↓) | 88.89 | 22.18M (0.56%↑) |
> | + $G_3B^\dagger$ | **84.23 (3.89%↑)** | 99.30 | 59.27 (10.85%↑) | 85.87 | - | - | 23.58M (6.94%↑) |
> | + $G_3U^\dagger$ | 84.06 (3.68%↑) | 99.19 | **60.18 (12.55%↑)** | 86.30 | - | - | 22.18M (0.56%↑) |
> | **ViT-Tiny** | 72.76 | 98.14 | 43.53 | 75.00 | **59.15** | 82.07 | 5.53M |
> | + $[G_3B]_{12}$ | 77.93 (7.11%↑) | 98.76 | 50.54 (16.10%↑) | 80.54 | 58.98 (0.29%↓) | 81.91 | 5.59M (1.08%↑) |
> | + $G_3B^\dagger$ | 77.81 (6.94%↑) | 98.76 | 48.33 (11.03%↑) | 78.68 | - | - | 6.29M (13.74%↑) |
> | + $G_3U^\dagger$ | **79.08 (8.69%↑)** | 98.91 | **49.33 (13.32%↑)** | 79.40 | - | - | 5.59M (1.08%↑) |
>
> Table 6 is also given here for the convenience of the reviewer:
>
> | Model | SVHN (Acc.@1) | Flowers 102 (Acc.@1) | STL-10 (Acc.@1) |
> | :--- | :--- | :--- | :--- |
> | **ViT-Base** | 97.74 | **92.24** | **97.26** |
> | + $[G_3B]_{12}$ | **97.81** | 91.97 | 96.65 |
> | **ViT-Small** | 97.48 | 91.46 | **96.09** |
> | + $[G_3B]_{12}$ | **97.56** | **91.59** | 95.79 |
> | **ViT-Tiny** | **96.69** | 84.21 | 93.20 |
> | + $[G_3B]_{12}$ | 96.40 | **85.53** | **93.35** |
>
> We sincerely thank the reviewer for suggesting experiments with positioning KArAt in different layers along with softmax, as it resulted in strategy *(i)*. In the initial layers KArAt does not help much as vision transformers learn coarse features in the initial layers. However, KArAt becomes extremely beneficial for the final layers due to high attention entropy. It is also an interesting idea to check for different grid sizes, $G$, and different hidden dimensions, $r$, in different layers. However, we are already using quite low $G$ and $r$ values throughout the network, and layer-wise tuning such values as hyperparameters would be impractical and time-consuming. Moreover, our ablations on hidden dimensions and grid sizes do not show significant accuracy improvements; see Sections C.4.2, C.4.3, C.4.4 in the Appendix.

---

### Official Review · Reviewer_TBHR · 2025-11-01

**Soundness:** 2
**Presentation:** 3
**Contribution:** 2
**Rating:** 4
**Confidence:** 4

**Summary:**

The authors present  a method to incorporate Kolmogorov-Arnold network into Vision Transformer. The prosed method is to replace softmax operation in self-attention with learnable operator based on KA network. The authors clain that the learnable operation may capture the token-to-token relationship better.

Since the full rank KAN operator would lead to memory-explosion, the authors propose a low rank approximation with Fourier basis.

The evaluation shows that the ViT tiny could achieve significant gain by changin the softmax with KAN.

**Strengths:**

1. This paper raises a question about fixed softmax operation in self-attention.
2. The results in ViT tiny model is promising.

**Weaknesses:**

1. First of all, this method does not scale properly. Even the performance with ViT-Base is disappointing.
2. Memory overhead even with low rank approximation is not justified.
3. As shown in Figure 3, the loss landscape becomes less smooth, which may results in training instability.

**Questions:**

1. How to overcome the scalability problem?
2. In case of standard ViT, the token-to-token relationship is captured by softmax and projection of learnable heads. In that sense, standard self-attention also  learns how to capture the relationship. Is KAN really better approach?
3. How to justify the memory overhead.

---

> ### Author Response · Authors · 2025-11-26
> **Rebuttal**
>
> ### 1. **Scalability of KArAt**
>
> The fundamental challenge of training KArAt via the modern deep learning toolkits in the present-day GPU and CUDA interface remains its scalability with larger overparameterized models. Although the system-level modification is beyond the scope of this paper, we discuss two potential solutions that can alleviate the curse of overparameterization in KArAt: *(i)* **parameter reduction by using KArAt only in a few layers**, and *(ii)* **improving the training instability by attention sparsification** in **Section 7 Improving KArAt’s Scalability**. We show that both the strategies improve scalability for ViT-Small and -Base, and in some cases improve performance in ViT-Tiny as well. Using these two strategies, for ViT-Tiny, -Small, -Base, we show maximums of *7.89%, 3.89%, 4.16% improvements on CIFAR-10, and 16.10%, 12.55%, 10.50% improvement on CIFAR-100, respectively, over traditional softmax attention.* **On ImageNet-1K, the accuracy drop remains less than a modest 0.78% and 0.77% on the ViT-Base and -Small respectively; see Table 5. This is a vast improvement from the results in Table 1 where Fourier KArAt was showing a performance drop of 6.68% compared to ViT-Base, and 3.89% compared to ViT-Small.**
>
> Additionally, in Table 6, we show that the strategy *(i)* with Fourier KArAt outperforms the traditional MHSA on SVHN for the ViT-Base and ViT-Small, on Flowers 102 for ViT-Small and Vit-Tiny, and on STL-10 for ViT-Tiny. These empirical results prove that properly guided strategies can improve scalability of KArAt in larger models; see our discussion in Section 7 of the recently updated version of the paper.
>
> Table 5 is given below for the convenience of the reviewer:
>
> | Model | CIFAR-10 Acc.@1 | CIFAR-10 Acc.@5 | CIFAR-100 Acc.@1 | CIFAR-100 Acc.@5 | ImageNet-1K Acc.@1 | ImageNet-1K Acc.@5 | Parameters |
> | :--- | :--- | :--- | :--- | :--- | :--- | :--- | :--- |
> | **ViT-Base** | 83.45 | 99.19 | 58.07 | 83.70 | **72.90** | 90.56 | 85.81M |
> | + $[G_3B]_{12}$ | 86.74 (3.94%↑) | 99.51 | 64.07 (10.33%↑) | 87.29 | 72.33 (0.78%↓) | 89.99 | 86.82M (1.18%↑) |
> | + $G_3B^\dagger$ | **86.92 (4.16%↑)** | 99.42 | 63.49 (9.33%↑) | 88.14 | - | - | 88.87M (3.57%↑) |
> | + $G_3U^\dagger$ | 84.82 (1.64%↑) | 99.24 | **64.17 (10.50%↑)** | 87.93 | - | - | 86.06M (0.29%↑) |
> | **ViT-Small** | 81.08 | 99.02 | 53.47 | 82.52 | **70.50** | 89.34 | 22.05M |
> | + $[G_3B]_{12}$ | 83.86 (3.43%↑) | 99.39 | 60.16 (12.51%↑) | 86.44 | 69.96 (0.77%↓) | 88.89 | 22.18M (0.56%↑) |
> | + $G_3B^\dagger$ | **84.23 (3.89%↑)** | 99.30 | 59.27 (10.85%↑) | 85.87 | - | - | 23.58M (6.94%↑) |
> | + $G_3U^\dagger$ | 84.06 (3.68%↑) | 99.19 | **60.18 (12.55%↑)** | 86.30 | - | - | 22.18M (0.56%↑) |
> | **ViT-Tiny** | 72.76 | 98.14 | 43.53 | 75.00 | **59.15** | 82.07 | 5.53M |
> | + $[G_3B]_{12}$ | 77.93 (7.11%↑) | 98.76 | 50.54 (16.10%↑) | 80.54 | 58.98 (0.29%↓) | 81.91 | 5.59M (1.08%↑) |
> | + $G_3B^\dagger$ | 77.81 (6.94%↑) | 98.76 | 48.33 (11.03%↑) | 78.68 | - | - | 6.29M (13.74%↑) |
> | + $G_3U^\dagger$ | **79.08 (8.69%↑)** | 98.91 | **49.33 (13.32%↑)** | 79.40 | - | - | 5.59M (1.08%↑) |
>
> Table 6 is also given here for the convenience of the reviewer:
>
> | Model | SVHN (Acc.@1) | Flowers 102 (Acc.@1) | STL-10 (Acc.@1) |
> | :--- | :--- | :--- | :--- |
> | **ViT-Base** | 97.74 | **92.24** | **97.26** |
> | + $[G_3B]_{12}$ | **97.81** | 91.97 | 96.65 |
> | **ViT-Small** | 97.48 | 91.46 | **96.09** |
> | + $[G_3B]_{12}$ | **97.56** | **91.59** | 95.79 |
> | **ViT-Tiny** | **96.69** | 84.21 | 93.20 |
> | + $[G_3B]_{12}$ | 96.40 | **85.53** | **93.35** |

---

> > ### Author Response · Authors · 2025-11-26
> > **Rebuttal --- continued**
> >
> > ### 2. **Why KArAt over softmax Attention**
> >
> > Stemming from Kolmogorov-Arnold Representation Theorem (Theorem 1 in our paper, proposed by Kolmogorov and Arnold in circa 1956), KANs with learnable activation functions can approximate complex functions and provide an interpretable alternative to MLPs [1, 2, 3]. In the same spirit, Theorem 2.1 in the original KAN paper states that if the target function admits a structure of the composition of $L$-KAN layers with smooth activation functions, then we can use an $L$-layer KAN (with the choice of B-Spline basis function) to approximate it well. This allows them to model complex relations that lie in the data distribution. The authors of KAN claimed that KANs can discover the need for a new function whose numerical behavior suggests it may be a Bessel function; see Figure 23 (d) in [1]. In fact, Figure 4(e) in [1] shows an example where KANs are very close to discovering a new equation.
> >
> > By using a similar argument, we hypothesize that KANs could be a better fit for approximating the underlying non-linear relationship among the interactions between image patches and can be used for modeling token interactions with learnable components. Hence, KArAt can be viewed as an upgrade to vanilla self-attention, offering adaptability and more freedom for modeling token interactions in an interpretable way. KArAt provides the freedom to choose from a multitude of simple, easy-to-interpret, and tunable functions whose linear combination can produce a large class of unknown functions, rather than a fixed and known function, softmax. The traditional softmax attention adds a non-linearity for modeling such interactions in a naive *“one-size-fits-all"* mechanism without any justification. This interpretability is directly reflected in KArAt’s attention maps; see Figures 4 and 11. Moreover, the benefits of learnability in KArAt are also directly reflected in the performance improvement shown in Tables 5 and 6; see our detailed discussion on performance gain in Section 7, Lines 520-531 of the updated paper.
> >
> > [1] Liu et al. "KAN: Kolmogorov-Arnold Networks." ICLR, 2025.
> >
> > [2] Liu et al. "KAN 2.0: Kolmogorov-Arnold Networks Meet Science." Preprint arXiv:2408.10205, 2024.
> >
> > [3] Yang and Wang. "Kolmogorov-Arnold Transformer." ICLR, 2025

---

> > > ### Author Response · Authors · 2025-11-26
> > > **Rebuttal --- continued**
> > >
> > > ### 3. **Justification of the Memory Overhead**
> > >
> > > Following the original Kolmogorov-Arnold representation theorem, our initial architecture in Section 3 involved a $N \times N$, full-rank KAN operator, $\phi$, that was impossible to train due to its extremely high number of parameters; see our discussion in Lines 203-214. *E.g.,* ViT-Tiny has 5.53M parameters and requires nearly 0.9 GB of GPU memory to train. Implementing the full rank learnable operators (Equation 1) in ViT-Tiny with B-splines of order 5 and grid size 10 and Fourier basis with grid size 10 increases the parameter count to 30.68M and 39.06M, respectively. We could not train the version with B-splines for its humongous memory requirements, and B-spline computations are non-parallelizable. The one with the Fourier basis is parallelizable, but it takes approximately **60 GB of GPU memory when computing with a batch of one image**. **Hence, calculations with the full-rank operator, $\phi$, are prohibitively expensive, regardless of the basis function and would make the training infeasible.** To remedy this, our proposed architecture uses a lower-dimensional KAN operator, $\hat{\phi}$, followed by a learnable linear projector $W$. However, present GPUs and CUDA functions are not optimized for KAN implementation as noted by many before us [3,4,5,6,7]. Due to this, although, our low rank design choice may not increase the parameters significantly (see Table 1), our proposed Fourier KArAt variants with the low-rank approximation still utilizes $2.5-3\times$ more GPU memory than the traditional softmax attention (Table 15); see our discussion in Lines 439-452. We note that an efficient system-level approach and low-level implementation are required for a better implementation of KArAt, which is beyond the scope of this work, and this cannot be considered as KArAt’s shortcomings. The goal of this work was to investigate if self-attention in ViTs can be learned and the benefits it can bring to visual understanding tasks. We believe the performance improvements in Table 5 and 6, and our detailed discussion on KArAt’s flexibility, interpretability and explainability over traditional softmax attention in Section 5.3 establish learnable attentions are beneficial over softmax attention and justify the memory overhead.
> > >
> > > [1] Liu et al. "KAN: Kolmogorov-Arnold Networks." ICLR, 2025.
> > >
> > > [2] Liu et al. "KAN 2.0: Kolmogorov-Arnold Networks Meet Science." Preprint arXiv:2408.10205, 2024.
> > >
> > > [3] Yang and Wang. "Kolmogorov-Arnold Transformer." ICLR, 2025
> > >
> > > [4] Ruijters and Thévenaz. "GPU Prefilter for Accurate Cubic B-spline Interpolation." The Computer Journal, 55(1):15–20, 2012.
> > >
> > > [5] Ruijters et al. "Efficient GPU-Based Texture Interpolation using Uniform B-splines." Journal of Graphics Tools, 13(4):61–69, 2008.
> > >
> > > [6] Xu et al. "FourierKAN-GCF: Fourier Kolmogorov-Arnold Network–An Effective and Efficient Feature Transformation for Graph Collaborative Filtering." Preprint arXiv:2406.01034, 2024.
> > >
> > > [7] Pal and Das. "Understanding the Limitations of B-Spline KANs: Convergence Dynamics and Computational Efficiency." NeurIPS Workshops, 2024.

---

### Author Response · Authors · 2025-11-26
**Official Comment to the Reviewers, AC and SAC**

Dear AC & Reviewers,

We thank the reviewers for their constructive feedbacks on our work and recognizing the potential and novelty of our work. No reviewer raised fundamental issues, novelty, or technical concerns about our work. The comments were focused on scalability and applicability of KArAt and why learnable attention is superior than the traditional softmax attention. **We were able to address all the comments and resolved the scalability issue of KArAt by performing extensive training of 30 new KArAt variants on CIFAR-10, CIFAR-100 and ImageNet-1K.** Below, we give a summary.

**Reviewer TBHR:**

- We address the question of scalability and performance on large ViTs with new experiments and results.

- We provided the explanation on why KArAt can be preferable over softmax for visual understanding tasks.

- We justify the memory overhead and explain our design choices.

**Reviewer ouZw:**

- We explained how we numerically justify the explainability of KArAt from their attention map visualizations.

- We provided new empirical data that addresses the scalability issue of KArAt.

**Reviewer aECS:**

- We have provided detailed explanations on the root cause of performance degradation in larger models, empirically validate the same, and show possible solutions.

- We discussed the efficiency aspect of the proposed learnable KArAt and provided justification behind our architecture design.

- We discussed the difference between traditional softmax attention and KArAt and pointed the benefits of KArAt over the softmax attention.

**Reviewer eokU:**

- We address the scalability problem with detailed empirical data.

- We discussed how KArAt offers better interpretability and explainability over traditional softmax attention and pointed out its other key benefits.


Overall, we added new experiments and result data to address the scalability issues which address the majority of the questions from the reviewers. The paper, including the appendix, is revised and updated. The new additions and modifications are done in **text color Blue**. The deletions are highlighted using **Red Strikethrough**. We modified **Section 1 Introduction**, **Section 6 Discussion**. We added a new Section, **Section 7 Improving KArAt’s Scalability**, and **Tables 5 and 6** in it. We moved the conclusion from Section 6 and added to **Section 8**.

We hope that reviewers will reconsider their ratings in light of the new empirical data and the justifications provided. We request that the AC facilitate further discussions. KArAt is the first-ever design of learnable attention in ViTs and we request the AC to consider our work as a novel research direction, instead of mere performance improvement.

Sincerely,

The authors

---

### Meta-Review · Area_Chair_moka · 2026-01-06

**Summary:**

Across all the reviewers, the scalability and generality of the proposed model, KArAt, is the major concern in terms of model performance degradation. While there are comments around the memory overhead, model parameters, visualisation, as well as explainability.

**Reviewer Concerns:**

As mentioned on the concerns, the most of the comments have been address well with more experiments and discussion. However, the scalability of this proposed model is still unconvincing and under validated as all the benefits are only validated on small scale datasets such as CIFAR10, SVH, etc, and the standard dataset ImageNet1K is not well supported.

**Reviewer Scores:**

While additional experiments have been conducted, this work is still not convincing in terms of scalability and generality. Using hybrid design could raise new questions about how well other types of combination would work, leaving this end open and not conclusive.

Also, the impact of this model could be restricted by the current implementation of hardwares.

Given these concerns, it is challenging to get the reviewers lift up the scores to be positive.

---

### Decision · Program_Chairs · 2026-01-26

Reject